# Multivariate stochastic volatility modeling of neural data

**Tung D Phan[†]\*, Jessica A Wachter[‡], Ethan A Solomon[§], Michael J Kahana[†]\***

University of Pennsylvania, Philadelphia, United States

**Abstract** Because multivariate autoregressive models have failed to adequately account for the complexity of neural signals, researchers have predominantly relied on non-parametric methods when studying the relations between brain and behavior. Using medial temporal lobe (MTL) recordings from 96 neurosurgical patients, we show that time series models with volatility described by a multivariate stochastic latent-variable process and lagged interactions between signals in different brain regions provide new insights into the dynamics of brain function. The implied volatility inferred from our process positively correlates with high-frequency spectral activity, a signal that correlates with neuronal activity. We show that volatility features derived from our model can reliably decode memory states, and that this classifier performs as well as those using spectral features. Using the directional connections between brain regions during complex cognitive process provided by the model, we uncovered perirhinal-hippocampal desynchronization in the MTL regions that is associated with successful memory encoding.
DOI: https://doi.org/10.7554/eLife.42950.001

**\*For correspondence:**
tungphan87@gmail.com (TDP);
kahana@psych.upenn.edu (MJK)

**Present address:** [†]Department of Psychology, University of Pennsylvania, Philadelphia, United States; [‡]Department of Finance, The Wharton School, University of Pennsylvania, Philadelphia, United States; [§]Department of Bioengineering, University of Pennsylvania, Philadelphia, United States

**Competing interests:** The authors declare that no competing interests exist.

## Introduction

Recent advances in neuroscience have enabled researchers to measure brain function with both high spatial and temporal resolution, leading to significant advances in our ability to relate complex behaviors to underlying neural signals. Because neural activity gives rise to electrical potentials, much of our knowledge concerning the neural correlates of cognition derive from the analyses of multi-electrode recordings, which yield a multivariate time series of voltage recorded at varying brain locations (denoted here as $y_t$). Such signals may be measured non-invasively, using scalp electroencephalography (EEG), or invasively, using subdural grids or intraparenchymal depth electrodes in human neurosurgical patients. In recent years, intracranially recorded (iEEG) signals have yielded detailed information on correlations between time-series measures and a wide range of behaviors including perception, attention, learning, memory, language, problem solving and decision making (*Jacobs and Kahana, 2010*).

Whereas other fields that grapple with complex multivariate time series have made effective use of parametric models such as economics and engineering (*Kim et al., 1998*; *Blanchard and Simon, 2001*; *West, 1996*), neuroscientists largely ceded early parametric approaches (e.g. linear autoregressive models) in favor of non-parametric spectral decomposition methods, as a means of uncovering features of neural activity that may correlate with behavior. A strength of these non-parametric methods is that they have enabled researchers to link fluctuations in iEEG signals to low-frequency neural oscillations observed during certain behavioral or cognitive states, such as slow-wave sleep (*Landolt et al., 1996*; *Chauvette et al., 2011*; *Nir et al., 2011*), eye closure (*Klimesch, 1999*; *Goldman et al., 2002*; *Laufs et al., 2003*; *Barry et al., 2007*) or spatial exploration (*Kahana et al., 2001*; *Raghavachari et al., 2001*; *Caplan et al., 2003*; *Ekstrom et al., 2005*; *Byrne et al., 2007*). High-frequency neural activity, which has also been linked to a variety of cognitive and behavioral states (*Maloney et al., 1997*; *Herrmann et al., 2004*; *Canolty et al., 2006*), is less clearly oscillatory, and may reflect asynchronous stochastic volatility of the underlying EEG signal (*Burke et al., 2015*).

Although spectral analysis methods have been used extensively in the neuroscience literature, they assume that there is unique information in each of a discrete set of frequency bands. The number of bands and frequency ranges used in these methods have been the subject of considerable controversy. Indeed *Manning et al. (2009)* have shown that broadband power often correlates more strongly with neuronal activity than does power at any narrow band. Also, non-parametric methods implicitly assume that the measured activity is observed independently during each observational epoch, and at each frequency, an assumption which is easily rejected in the data, which show strong temporal autocorrelation as well as correlations among frequency bands (*von Stein and Sarnthein, 2000*; *Jensen and Colgin, 2007*; *Axmacher et al., 2010*). Moreover, non-parametric methods are typically applied to EEG signals in a univariate fashion that neglects the spatial correlational structure. By simultaneously modeling the spatial and temporal structure in the data, parametric models confer greater statistical power so long as they are not poorly specified.

Parametric methods have been applied to various types of multivariate neural data including EEG (*Hesse et al., 2003*; *Dhamala et al., 2008*; *Bastos et al., 2015*), magnetoencephalography (MEG) (*David et al., 2006*), functional magnetic resonance imaging (FMRI) (*Roebroeck et al., 2005*; *Goebel et al., 2003*; *David et al., 2008*; *Luo et al., 2013*), and local field potentials (LFP) (*Brovelli et al., 2004*). These methods typically involve fitting vector autoregressive (VAR) models to multivariate neural data that are assumed to be stationary in a specific time interval of interest. The regression coefficient matrix derived from the VAR models can be used to study the flow of information between neuronal regions in the context of Granger causality (G-causality). Neuroscientists have used Gaussian VAR models to study the *effective connectivity* (directed influence) between activated brain areas during cognitive and visuomotor tasks (*Zhou et al., 2009*; *Deshpande et al., 2009*; *Graham et al., 2009*; *Roebroeck et al., 2005*). Although VAR models and G-causality methods have been argued to provide useful insights into the functional organization of the brain, their validity relies upon the assumptions of linearity and stationarity in mean and variance of the neural data. When one of these assumptions is violated, the conclusions drawn from a G-causality analysis will be inconsistent and misleading (*Seth et al., 2015*). One of the most common violations by EEG signals is the assumption of variance-stationarity (*Wong et al., 2006*). Therefore, in the present work, we adopt a stochastic volatility approach in which the non-stationary variance (also known as volatility) of the neural time series is assumed to follow a stochastic process. Such models have been extremely useful in the analyses of financial market data which, like neural data, exhibits high kurtosis (*Heston, 1993*; *Bates, 1996*; *Barndorff-Nielsen and Shephard, 2002*).

We propose a multivariate stochastic volatility (MSV) model with the aim of estimating the time-varying volatility of multivariate neural data and its spatial correlational structure. The MSV model assumes that the volatility series of iEEG signals follows a latent-variable vector-autoregressive process, and it allows for the lagged signals of different brain regions to influence each other by specifying a full persistent matrix (typically assumed to be diagonal) in the VAR process for volatility.

We employed a Bayesian approach to estimate the latent volatility series and the parameters of the MSV model using the forward filtering backward sampling and Metropolis Hastings algorithms. We validated the MSV model in a unique dataset comprising depth-electrode recordings from 96 neurosurgical patients. These patients volunteered to participate in a verbal recall memory task while they were undergoing clinical monitoring to localize the epileptogenic foci responsible for seizure onset. Our analyses focused on the subset of electrodes ($n = 718$) implanted in medial temporal lobe (MTL) regions, including hippocampus, parahippocampal cortex, entorhinal cortex and perirhinal cortex. We chose to focus on these regions given their prominent role in the encoding and retrieval of episodic memories (*Davachi et al., 2003*; *Kirwan and Stark, 2004*; *Kreiman et al., 2000*; *Sederberg et al., 2007*).

We show that the MSV model, which allows for interactions between regions, provides a substantially superior fit to MTL recordings than univariate stochastic volatility (SV) models. The implied volatility in these models positively correlates with non-parametric estimates of spectral power, especially in the gamma frequency band.

We demonstrate the utility of our method for decoding cognitive states by using a penalized logistic regression classifier trained on the implied volatility data across MTL electrodes to predict which studied items will be subsequently recalled. We find that the MSV-derived features outperform spectral features in decoding cognitive states, supporting the value of this model-based time-series analysis approach to the study of human cognition. Furthermore, using the MSV model to

construct a directional MTL connectivity network, we find that significant bidirectional desynchroni-
zation between the perirhinal cortex and the hippocampus predicts successful memory encoding.

## Multivariate Stochastic Volatility models for iEEG

### Volatility of iEEG is stochastic

Previous studies have shown that variance of EEG recordings is time-varying (*Wong et al., 2006*;
*Galka et al., 2010*). *Klonowski et al. (2004)* demonstrated a striking similarity between the time-
series of the Dow Jones index during economic recessions (big crashes) and the timeseries of iEEG
during epileptic seizures. These timeseries typically possess 'big spikes' that are associated with
abrupt changes in the variance of the measured signal. There are two main approaches for modeling
the time-varying variance commonly known as volatility in the econometrics literature, both of which
assume that volatility is a latent autoregressive process, that is the current volatility depends on its
previous values. The first approach is the class of autoregressive conditional heteroscedastic (ARCH)
models developed by *Engle (1982)* and the generalized autoregressive conditional heteroscedastic
(GARCH) models extended by *Bollerslev (1986)*. The ARCH/GARCH models assume that the cur-
rent volatility is a deterministic function of the previous volatilities and the information up to the cur-
rent time. The second approach is the class of stochastic volatility models proposed by
*Heston (1993)*, which assume that volatility is non-deterministic and follows a random process with
a Gaussian error term. GARCH-type models have been popular in the empirical research community
since the 1980's due to their computational attractiveness. Inference of GARCH models is usually
performed using maximum likelihood estimation methods (*Engle and Bollerslev, 1986*; *Nel-
son, 1991*). Until the mid 1990s, stochastic volatility models had not been widely used due to the
difficulty in estimating the likelihood function, which involves integrating over a random latent pro-
cess. With advances in computer technology, econometricians started to apply simulation-based
techniques to estimate SV models (*Kim et al., 1998*; *Chib et al., 2002*; *Jacquier et al., 1994*).
Despite their computational advantages, the GARCH model assumes a deterministic relationship
between the current volatility and its previous information, making it slow to react to instantaneous
changes in the system. In addition, *Wang (2002)* showed an asymptotic nonequivalance between
the GARCH model and its diffusion limit if volatility is stochastic. Therefore, we employ a more gen-
eral stochastic volatility approach to analyze iEEG signals.

To motivate the discussion of the MSV model, we first examine the distributional property of vola-
tility. Previous econometrics literature has shown that volatilities of various financial timeseries can
be well approximated by log-normal distributions such as the Standard and Poor 500 index
(*Cizeau et al., 1997*), stock returns (*Andersen, 2001a*), and daily foreign exchange rates
(*Andersen et al., 2001b*). The volatilities of these financial measures are found to be highly right-
skewed and their logarithmic volatilities are approximately normal. Volatility of iEEG also exhibits
this log-normality property. To demonstrate this property, we calculate and plot the density of the
empirical variance of a sample iEEG timeseries using a rolling variance of window size 20 (*Hull, 2009*,
chapter 17). *Figure 1* demonstrates the distribution of the empirical volatility timeseries of the
detrended (after removing autoregressive components) raw iEEG signals. The distribution is right-
skewed and can be well approximated by a log-normal distribution. Due to this log-normality prop-
erty, the SV approach typically models the logarithm of volatility instead of volatility itself. The next
section describes the multivariate stochastic volatility model for iEEG data, which is a generalized
version of the SV model that accounts for the interactions between recording brain locations.

### The model

Stochastic volatility models belong to a wide class of non-linear state-space models that have been
extensively used in financial economics. There has been overwhelming evidence of non-stationarity
in the variance of financial data (*Black et al., 1972*) and much effort has been made to model and
understand the changes in volatility in order to forecast future returns, price derivatives, and study
recessions, inflations and monetary policies (*Engle and Patton, 2001*; *Cogley and Sargent, 2005*;
*Blanchard and Simon, 2001*). There is by now a large literature on stochastic volatility models and
methods for estimating these models either by closed-form solutions (*Heston, 1993*; *Heston and
Nandi, 2000*) or by simulation (*Harvey and Shephard, 1996*; *Kim et al., 1998*; *Omori et al., 2007*).
Under the stochastic volatility framework, the variance (or its monotonic transformation) of a time

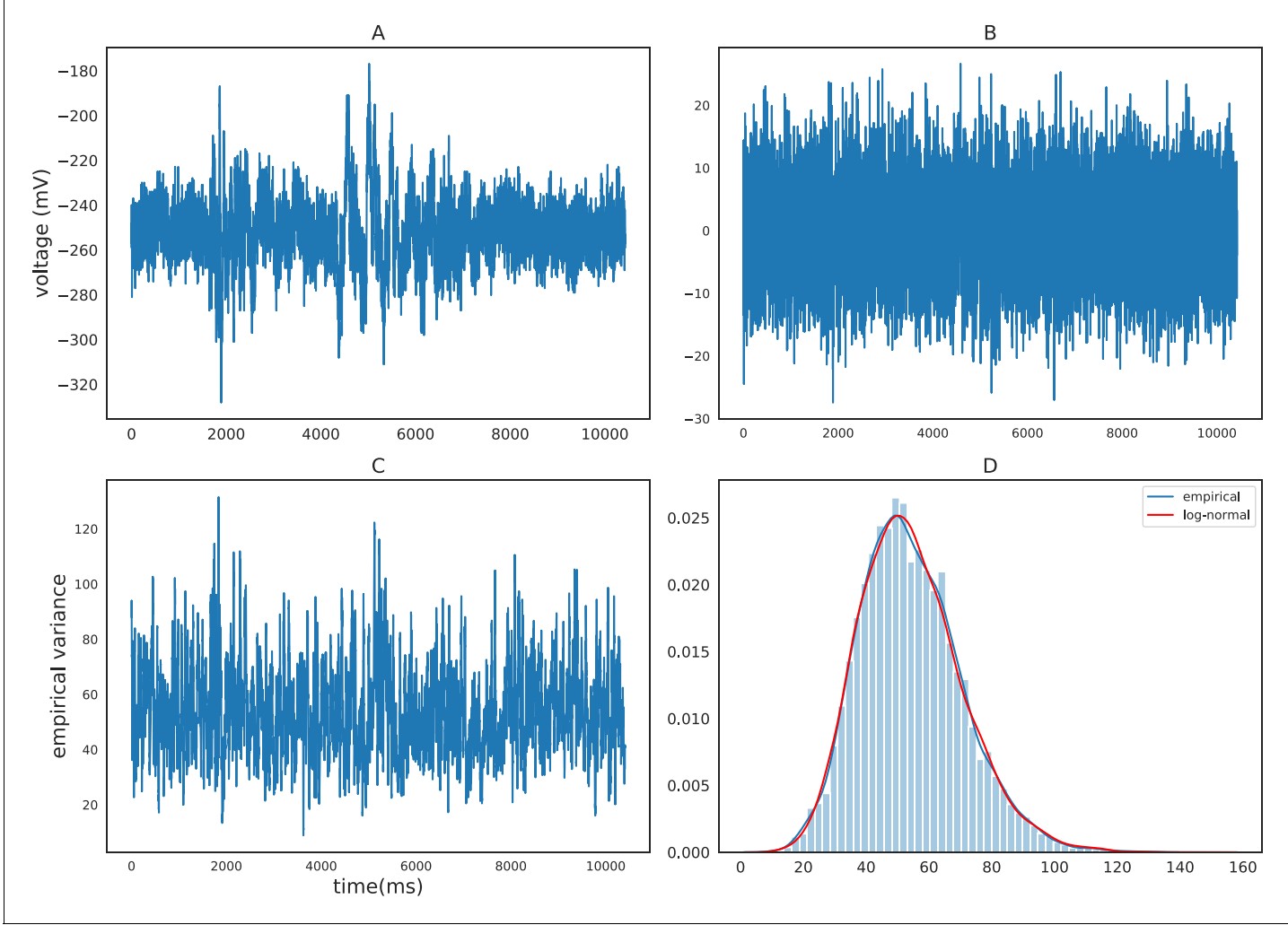

**Figure 1.** Empirical characteristics of iEEG. (**A**) Sample raw iEEG time series during a resting state (count down non-task) period for subject R1240T. (**B**) Detrended iEEG timeseries after removing autoregressive components. (**C**) Empirical variance timeseries calculated using a rolling-window of size 20. (**D**) Distribution of empirical variance with the blue curve showing the estimated empirical density using kernel density estimation and the red curve showing the best-fitting log-normal density to the data.

DOI: https://doi.org/10.7554/eLife.42950.002

series is assumed to be a latent process, which is typically assumed to be autoregressive. Latent process models have also been widely applied to the neuroscience domain to model neuronal spiking activity using point processes (*Macke et al., 2011*; *Smith and Brown, 2003*; *Eden et al., 2004*), to study motor cortical activity using a state-space model (*Wu et al., 2006*), etc. These models have provided many insights into the mechanisms underlying cognitive processes. GARCH-type models have also been applied to EEG signals to study the transition into anesthesia in human, sleep stage transitions in sheep, and seizures in epileptic patients (*Galka et al., 2010*; *Mohamadi et al., 2017*). However, the applications of the GARCH models in the neuroscience literature remain on a small-scale, which focus on individual recording locations and neglect the connectivity among different regions of the brain. In this study, we provide a systematic way to study volatility of iEEG signals using a large iEEG dataset from the MTL region during a verbal memory task as a medium to illustrate how volatility and its connectivity network among MTL subregions can provide insights into the understanding of cognitive processes.

Following *Harvey et al. (1994)*, we model the multivariate latent volatility process of the iEEG signals in the MTL region to follow a vector autoregressive model. The original model assumes that the coefficient matrix is diagonal, that is the past activity of one region does not have any influence

on the others. In many financial applications, it is convenient to make this diagonality assumption to reduce the number of parameters of the MSV model, otherwise, a very large amount of data would be required to reliably estimate these parameters. We generalize the MSV model to allow for a full coefficient matrix in order to study the directional connections between different subregions in the MTL. This generalization is feasible due to high-temporal-resolution neural time series collected using the intracranial electroencephalography. Let $\mathbf{y}_t = (y_{1,t}, \cdots, y_{J,t})$ be a multivariate iEEG time-series recordings at $J$ electrodes at time $t$. We model $y_{j,t}$, $1 \leq j \leq J$, as follows:

$$y_{j,t} = \exp(\frac{x_{j,t}}{2})\epsilon_{j,t}^y, \tag{1}$$

and

$$x_{j,t} - \mu_j = \sum_{k=1}^{J} \beta_{j,k}(x_{k,t-1} - \mu_k) + \epsilon_{j,t}^x, \tag{2}$$

where the error terms follow multivariate normal distributions: $\epsilon_t^y = (\epsilon_{1,t}^y, \cdots, \epsilon_{J,t}^y) \sim \mathcal{MVN}(\mathbf{0}, I_J)$, $\epsilon_t^x = (\epsilon_{1,t}^x, \cdots, \epsilon_{J,t}^x) \sim \mathcal{MVN}(\mathbf{0}, \Sigma)$ denotes the identity matrix of rank $J$, and $\Sigma = diag(\sigma_1^2, \cdots, \sigma_J^2)$ is assumed to be diagonal. That is, $\{y_{j,t}\}$ is a time series whose conditional log-variance (log-volatility), $\{x_{j,t}\}$, follows an AR(1) process that depends on its past value and the past values of other electrodes. The series $\{y_{1,t}\}_{t=1}^T, \cdots, \{y_{J,t}\}_{t=1}^T$ are assumed to be conditionally independent given their log-volatility series $\{x_{1,t}\}_{t=1}^T, \cdots, \{x_{J,t}\}_{t=1}^T$. The coefficient $\beta_{j,k}$ models how the past value of channel $k$ affects the current value of channel $j$ and $\mu_k$ is the unconditional average volatility at channel $k$. We can rewrite *Equation 2* in a matrix form

$$\mathbf{x}_t - \mu = \beta(\mathbf{x}_{t-1} - \mu) + \epsilon_t^x, \tag{3}$$

where $\mathbf{x}_t = (x_{1,t}, \cdots, x_{J,t})$, $\mu = (\mu_1, \cdots, \mu_J)$, and $\beta(j,k) = \beta_{j,k}$. The vector error terms $\epsilon_t^y$ and $\epsilon_t^x$ are assumed to be independent. The parameters in the system above are assumed to be unknown and need to be estimated.

Following a Bayesian perspective, we assume that the parameters are not completely unknown, but they follow some prior distributions. Then, using the prior distributions and the information provided by the data, we can make inferences about the parameters from their posterior distributions.

## Priors and estimation method

We specify prior distributions for the set of parameters $\theta = (\mu, \beta, \Sigma)$ of the MSV model. The mean vector $\mu$ follows a flat multivariate normal distribution $\mu \sim \mathcal{MVN}(0, 1000 I_J)$. Each entry of the persistence matrix $\beta_{i,j} \in (-1, 1)$ is assumed to follow a beta distribution, $(\beta_{i,j} + 1)/2 \sim Beta(20, 1.5)$ (*Kim et al., 1998*). The beta prior distribution ensures that the entries of the persistent matrix are between $-1$ and $1$, which guarantees the stationarity of the volatility process. For volatility of volatility, we utilize a flat gamma prior, $\sigma_j \sim \Gamma(1/2, 1/2 \times 10)$ (*Kastner and Frühwirth-Schnatter, 2014*), which is equivalent to $\pm\sqrt{\sigma_j^2} \sim N(0, 10)$. We estimated the latent volatility processes and the parameters of the MSV model using a Metropolis-within-Gibbs sampler (*Kim et al., 1998*; *Omori et al., 2007*; *Kastner and Frühwirth-Schnatter, 2014*) (see Appendix 1 for derivation and 2 for discussion of parameter identification). Here, we choose hyperparameters which provide relatively flat prior distributions. In addition, due to a large amount of data ($\approx 1$ million time points per model for each subject) that was used to estimate the MSV model, the choice of the hyperparameters has little effect on the posterior estimates of the parameters in the MSV model.

## Applications to verbal-free recall task

We analyzed the behavioral and electrophysiological data of 96 subjects implanted with MTL subdural and depth electrodes during a verbal free recall memory task (see Materials and methods section for details), a powerful paradigm for studying episodic memory. Subjects learned 25 lists of 12 unrelated words presented on a screen (encoding period) separated by 800–1200 ms interstimulus intervals (ISI), with each list followed by a short arithmetic distractor task to reduce recency effects (subjects more likely to recall words at the end of the list). During the retrieval period, subjects

recalled as many words from the previously studied list as possible, in any order (*Figure 2*). In this paper, we focused our analyses on the MTL regions that have been implicated in episodic memory encoding (*Squire and Zola-Morgan, 1991*; *Solomon et al., 2019*; *Long and Kahana, 2015*). To assess a particular effect across subjects, we utilized the maximum a posteriori (MAP) estimate by taking the posterior mean of the variable of interest (whether it be the volatility time series $x_t$ or the regression coefficient matrix $\beta$) (*Stephan et al., 2010*).

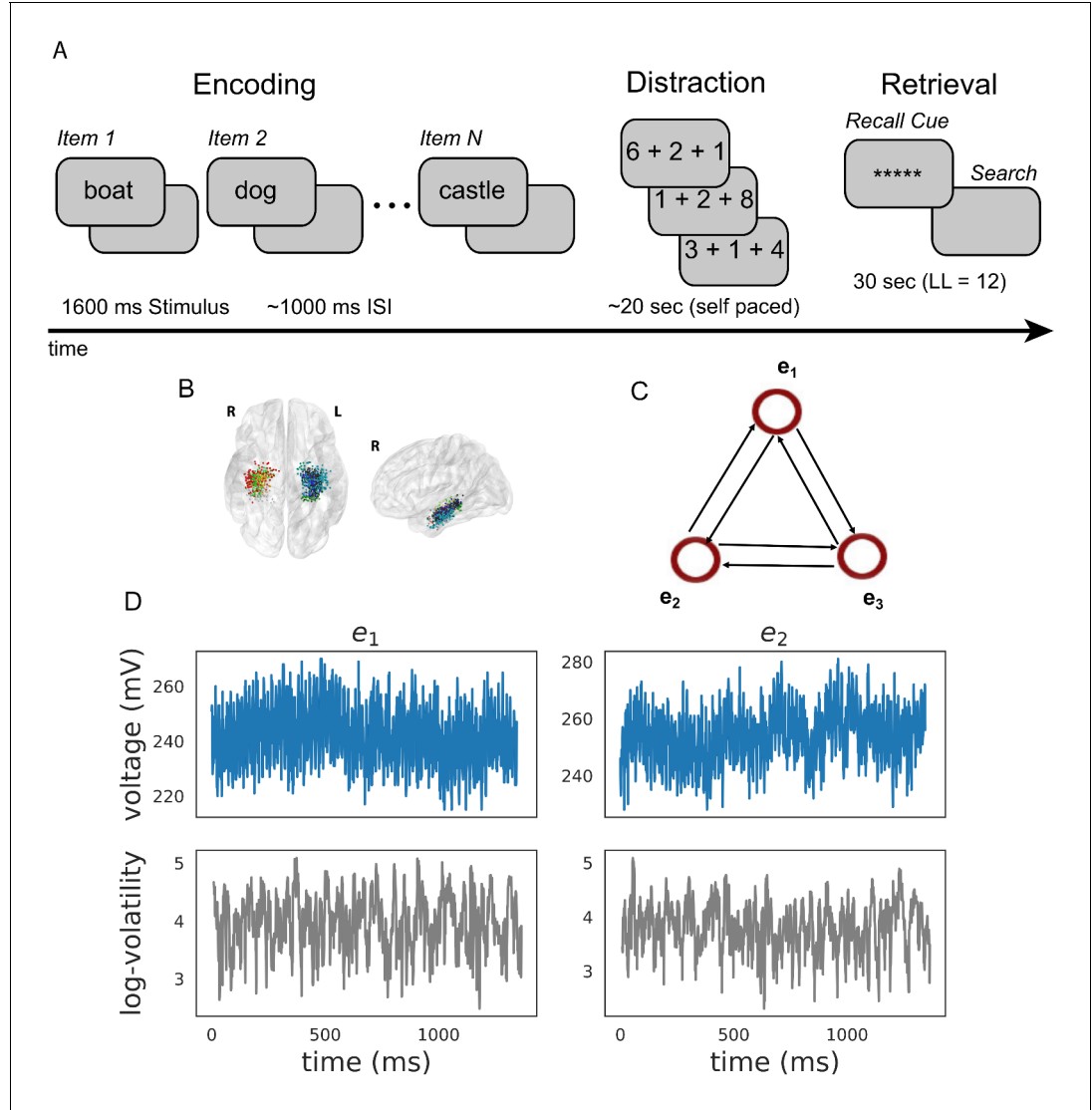

**Figure 2.** Task design and analysis. (**A**) Subjects performed a verbal free-recall task which consists of three phases: (1) word encoding, (2) math distraction, and (3) retrieval. (**B**) 96 Participants were implanted with depth electrodes in the medial temporal lobe (MTL) with localized subregions: CA1, CA3, dentate gyrus (DG), subiculum (Sub), perirhinal cortex (PRC), entorhinal cortex (EC), or parahippocampal cortex (PHC). (**C**) To construct a directional connectivity network, we applied the MSV model to brain signals recorded from electrodes in the MTL during encoding. We analyzed the 1.6 s epochs during which words were presented on the screen. The network reflects directional lag-one correlations among the implied volatility timeseries recorded at various MTL subregions. (**D**) The upper row shows a sample of an individual patient's raw voltage timeseries (blue) recorded from two electrodes during a word encoding period of 1.6 s, and the lower row shows their corresponding implied volatility timseries (gray) estimated using the MSV model.

DOI: https://doi.org/10.7554/eLife.42950.003

## Model comparison

To establish the validity of the MSV model, we compared its performance to that of univariate stochastic volatility models (equivalent to setting all the off-diagonal entries of the matrix $\beta$ in *Equation 2* to 0) in fitting iEEG data. We applied the MSV model to the multivariate neural data combined across encoding periods (regardless of whether the word items were later recalled) and SV models to datasets of individual electrodes with the assumption that the parameters of these models were shared across encoding events. We utilized the deviance information criterion (DIC) (*Spiegelhalter et al., 2002*; *Gelman et al., 2014*) considered to be a Bayesian analogue of the Akaike information criterion (AIC) to evaluate the performance of the models. The DIC consists of two components: the negative log-likelihood, $\bar{D} = \mathbb{E}_{\theta, \mathbf{x} \mid \mathbf{y}}[-2 \log P(\mathbf{y} \mid \theta, \mathbf{x})]$, which measures the goodness-of-fit of the model and the effective number of parameters, $p_D = \bar{D} - D(\bar{\theta}, \bar{\mathbf{x}}) = \mathbb{E}_{\theta, \mathbf{x} \mid \mathbf{y}}[-2 \log P(\mathbf{y} \mid \theta, \mathbf{x})] + 2 \log P(\mathbf{y} \mid \bar{\theta}, \bar{\mathbf{x}})$, which measures the complexity of the model (see Appendix 5 for details on how to compute DIC for the MSV model). Where $\bar{\theta}$ and $\bar{\mathbf{x}}$ denote the posterior means of the latent volatility series and the parameters of the MSV model. The DIC balances the trade-off between model fit and model complexity. Models with smaller DICs are preferred. To account for the varying amount of data each subject had, we averaged the DIC by the number of events and electrodes. We found the MSV model to have a consistently lower DIC value than the SV model with a mean difference of 23 ($\pm$5.9 SEM). This indicates that the MSV model is approximately more than 150 times as probable as the SV models (*Kass and Raftery, 1995*), suggesting that the MSV model is a more appropriate model for iEEG data.

## Relation to spectral power

We next analyzed the relation between volatility and spectral power (see Materials and methods) over a wide range of frequencies, from 3 to 180 Hz with 1 Hz steps). For each subject, we computed the correlation between volatility and spectral power for each encoding event and then averaged these correlations across all events. Since spectral powers of neighboring frequencies exhibit high correlations, we utilized a Gaussian regression model (*Rasmussen, 2004*) to estimate the correlation between volatility and spectral power as a function of frequency, which allows for a non-linear relation. *Figure 3* indicates that the correlation between volatility and spectral power is significantly positive across the spectrum and increasing in frequency. This illustrates the broadband nature of the volatility measure, but also suggests that volatility may more closely relate to previous neuroscientific findings observed for high-frequency as compared with low-frequency activity. Having established that the MSV model outperforms the more traditional SV approach, and having shown that the implied volatility of the series reliably correlates with high-frequency neural activity, we next asked whether we can use the model-derived time series of volatility to predict subjects' behavior in a memory task.

## Classification of subsequent memory recall

Extensive previous work on the electrophysiological correlates of memory encoding has shown that spectral power, in both the low-frequency (4–8 Hz) theta band and at frequencies about 40 Hz (so-called gamma activity), reliably predicts which studied words will be subsequently recalled or recognized (*Sederberg et al., 2003*). Here, we ask whether the implied volatility derived from the MSV model during word encoding can also reliably predict subsequent recall. To benchmark our MSV findings, we conducted parallel analyses of wavelet-derived spectral power at frequencies ranging between 3 and 180 Hz. To aggregate across MTL electrodes within each subject we applied an L2-penalized logistic regression classifier using features extracted during the encoding period to predict subsequent memory performance (*Ezzyat et al., 2017*; *Ezzyat et al., 2018*). To estimate the generalization of the classifier, we utilized a nested cross-validation procedure in which we trained the model on $N - 1$ sessions using the optimal penalty parameter selected via another inner cross-validation procedure on the same training data (see Appendix 6 for details). We then tested the classifier on a hold-out session collected at a different time. We computed the receiver operating characteristic (ROC) curve, relating true and false positives, as a function of the criterion used to assign regression output to response labels (see Appendix 6 for illustrations of ROC curves). We then use the AUC metric (area under the ROC curve) to characterize model performance. In order to perform a nested cross-validation procedure, we focused on 42 subjects (out of 96 subjects with at least two

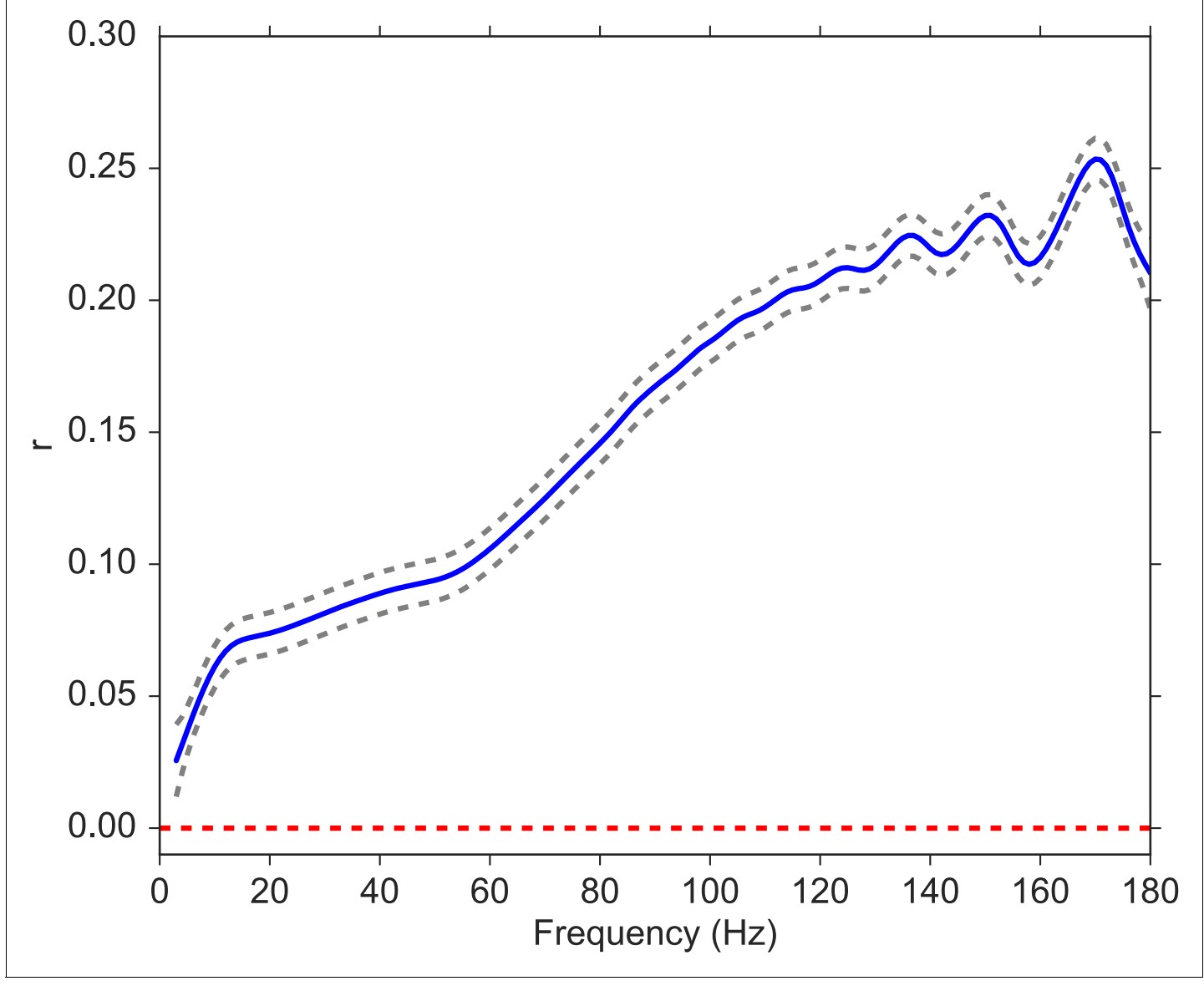

**Figure 3.** Correlation between volatility and spectral power over a frequency range from 3 to 180 Hz. We fit a Gaussian process model to estimate the functional form of the correlation function between volatility and spectral power (solid blue line). The 95% confidence bands were constructed from 96 subjects (dashed gray lines). The red line shows the null model. We observe a significantly positive correlation between volatility and spectral power, and the correlation increases with frequency.

DOI: https://doi.org/10.7554/eLife.42950.004

sessions) with at least three sessions of recording. We find that MSV-model implied volatility during item encoding reliably predicts subsequent recall, yielding an average AUC of 0.53 (95% CI, from 0.51 to 0.55). AUCs reliably exceeded chance levels in 72 percent of subjects (30 out of 42 subjects who contributed at least 3 sessions of data). *Figure 4* compares these findings against results obtained using wavelet-derived power. Here we see that implied volatility does as well as, or better than, spectral measures at nearly all frequencies. In order to capture the correlation between spectral powers (thus their corresponding classifiers' performances), we fit a Gaussian regression model to test the functional form of ΔAUC. We find that the ΔAUC function is significantly different from the 0 function ($\chi^2_{11} = 42$, P < $10^{-5}$) (*Benavoli and Mangili, 2015*), which indicates that on average volatility performs significantly better than spectral power in predicting subsequent memory recall.

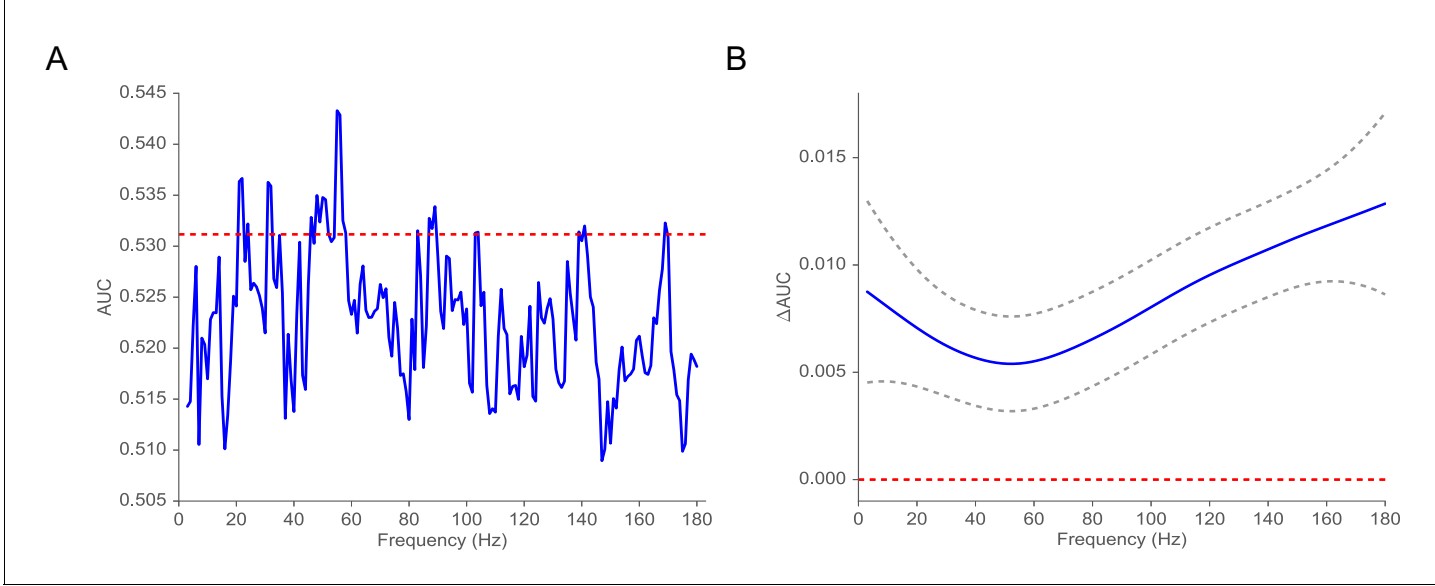

**Figure 4.** Classification of subsequent memory recall. (A) Average AUC of the classifier trained on spectral power across 42 subjects with at least three sessions of recording (blue). The red line indicates the average AUC of the classifier trained on volatility. (B) ΔAUC = $AUC_{vol} - AUC_{power}$ as a function of frequency estimated by using a Gaussian regression model (dashed gray lines indicate 95% confidence bands). The red line shows the null model. We observe that the classifier trained on volatility performs at least as well as the one trained on spectral power across the frequency spectrum. We find that functional form of ΔAUC is significantly different from the function ($\chi^2_{11} = 42$, P < $10^{-5}$) using a Gaussian process model, suggesting that the difference in performance between the volatility classifier and the spectral power classifier is significant.

DOI: https://doi.org/10.7554/eLife.42950.005

## Directional connectivity analysis

Having established that volatility is predictive of subsequent memory recall, we now seek to identify directional connections between MTL subregions that are related to successful memory encoding. To investigate the intra-MTL directional connectivity patterns that correlate with successful memory encoding, we utilize a subsequent memory effect (SME) paradigm in which we compare the MTL directional connectivity patterns (regression coefficient matrix β) associated with recalled (R) word items to those associated with non-recalled (NR) items. The SME paradigm has been widely used in the memory literature to study neural correlates (typically spectral power in a specific frequency band) that predict successful memory formation (*Sederberg et al., 2003*; *Long et al., 2014*; *Burke et al., 2014*). The intra-MTL connectivity SME was constructed using the following procedure. First, we partitioned the word items into recalled and non-recalled items offline. Using the MSV model, we constructed an intra-MTL connectivity network for each memory outcome. We compared the distribution of the elements of these matrices across subjects. For the analysis, we considered four subregions of the MTL: hippocampus (Hipp), entorhinal cortex (EC), perirhinal cortex (PRC), and parahippocampal cortex (PHC). Each MTL subregion contains a different number of recording locations depending on the subject's electrode coverage. We then computed the contrast between the two intra-MTL networks corresponding to recalled and non-recalled items for each ordered pair of subregions excluding the ones with fewer than 10 subjects contributing to the analysis. To compute the directional connectivity from region $I$ to region $J$, we took the average of the lag-one 'influences' that electrodes in region $I$ have on electrodes in region $J$, where $|I|$ denotes the number of electrodes in region $I$. We then computed the contrast between the two connectivity networks associated with recalled and non-recalled items: $\Delta_{I \to J} = C^R_{I \to J} - C^{NR}_{I \to J}$. Finally, we averaged the contrast for each ordered pair of MTL subregions across sessions within a subject. From now on, we refer to this contrast as simply connectivity network.

*Figure 5* illustrates the intra-MTL connectivity SME for the left and right hemispheres. Directed connections between the left hippocampus and the left PRC reliably decrease (false-discovery-rate-corrected) during successful memory encoding ($\Delta_{Hipp \to PRC} = -0.04, t_{47} = -3.49$, adj. P <0.01 and $\Delta_{PRC \to Hipp} = -0.06, t_{47} = -2.66$, adj. P <0.05). The difference between the directional connections

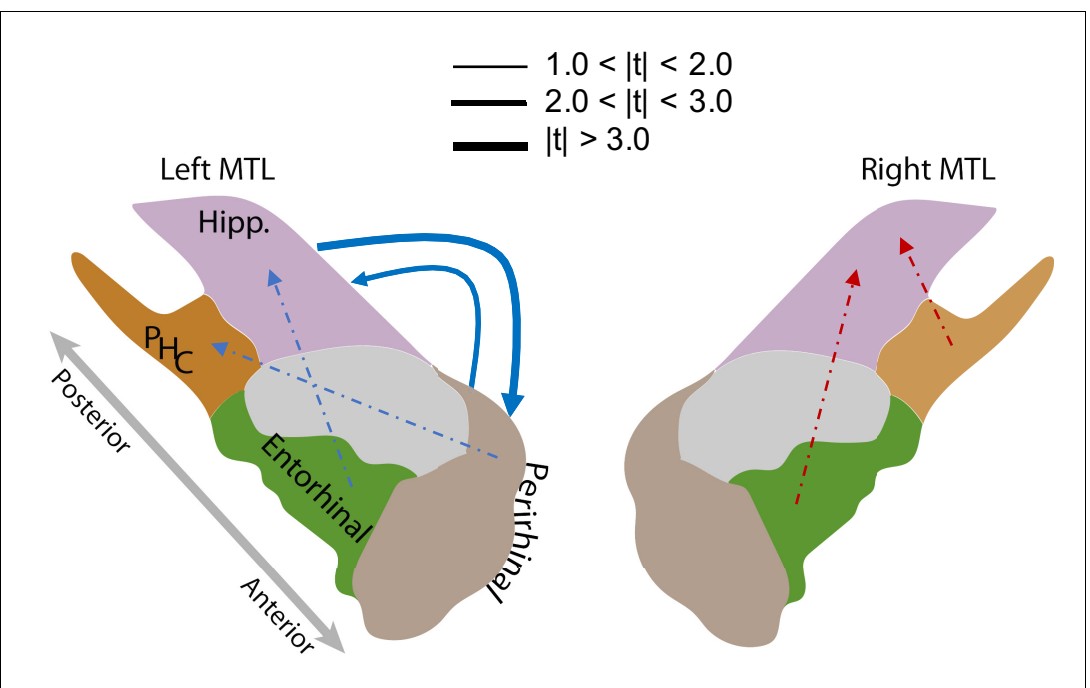

**Figure 5.** MTL directional connectivity network. The MTL electrodes were divided into four subregions: hippocampus (Hipp.), parahippocampal cortex (PHC), entorhinal cortex (EC), and perirhinal cortex (PRC). The directional connectivity from region I to region J, $C_{I \to J} = \frac{1}{|I||J|} \sum_{i \in I, j \in J} \beta_{ij}$, was calculated by averaging the entries of the sub-matrix of the regression coefficient matrix $\beta$, whose rows and columns correspond to region I and J respectively. We computed the contrast between the directional connectivity of recalled and non-recalled events: $\Delta_{I \to J} = C_{I \to J}^{R} - C_{I \to J}^{NR}$ for each subject. Solid lines show significant (FDR-corrected) connections between two regions and dashed lines show trending but insignificant connections. Red indicates positive changes and blue indicates negative changes. The directional connectivity from Hipp. to PRC is significant (adj. P < 0.01) and the reverse directional connectivity is also significant (adj. P < 0.05).

DOI: https://doi.org/10.7554/eLife.42950.006

between these two regions is not significant ($t_{47}$=0.53, p=0.60). The decreases in the bi-directional connections within the left MTL are consistent with the findings in *Solomon et al. (2017)* which noted memory-related decreases in phase synchronization at high frequencies. We did not, however, find any other significant directional connections among the remaining regions (*Figure 5*, *Appendix 7—tables 1* and *2*).

## Discussion

The ability to record electrophysiological signals from large numbers of brain recording sites has created a wealth of data on the neural basis of behavior and a pressing need for statistical methods suited to the properties of multivariate, neural, time-series data. Because neural data strongly violate variance-stationarity assumptions underlying standard approaches, such as Granger causality (*Stokes and Purdon, 2017*), researchers have generally eschewed these model-based approaches and embraced non-parameter data analytic procedures. The multivariate stochastic volatility framework that we propose allows for non-stationary variance in the signals. This framework allows us to explicitly model the time-varying variance of neural signals. Similar stochastic volatility models have been used extensively in the financial economics literature to characterize a wide range of phenomena.

The MSV models proposed in this paper provide a new framework for studying multi-channel neural data and relating them to cognition. Using a large MTL intracranial EEG dataset from 96 neurosurgical patients while performing a free-recall task, we found that volatility of iEEG timeseries is correlated with spectral power across the frequency spectrum and the correlation increases with frequency. To further test the ability of the MSV model to link iEEG recordings to behavior, we asked

whether MTL volatility features during encoding can predict subsequent memory recall as well as spectral power features. Our findings indicate that volatility features significantly outperform the spectral features in decoding memory process in the human brain, suggesting that volatility can serve as a reliable measure for understanding cognitive processes.

A key strength of the MSV approach is its ability to identify directed interactions between brain regions without assuming variance-stationarity of neural signals. We therefore used this approach to determine the directional connections between MTL subregions that correlate with successful memory encoding. Using the regression coefficient matrix of the multivariate volatility process, we found that periods of decreased connectivity in the volatility network among MTL subregions predicted successful learning. Specifically, we found that the hippocampus and the perirhinal cortex in the left hemisphere desynchronize (exerting less influence on one another) during successful learning, which is consistent with the late-phase gamma decoupling noted in *Fell et al. (2001)*. A more recent study by *Solomon et al. (2019)* also examined the association between intra-MTL connectivity and successful memory formation using phase-based measures of connectivity. *Solomon et al. (2019)* noted intra-MTL desynchronization at high frequencies during successful memory formation, aligning with the finding here that the volatility network tended to desynchronize. Furthermore, Solomon, et al. found broad increases in low-frequency connectivity, which did not appear to be captured by our stochastic model. This suggests that volatility features reflect neural processes that are also captured by high-frequency phase information.

We further noted more statistically reliable changes in volatility networks in the left MTL compared to the right. This result is in line with a long history of neuroanatomical, electrophysiological, and imaging studies (e.g. *Ojemann and Dodrill, 1985*; *Kelley et al., 1998*) that found an association between verbal memory and the left MTL. It is possible that the verbal nature of our memory task specifically engaged processing in the left MTL, resulting in a lateralization of observed volatility phenomena.

Prior studies implicate the perirhinal cortex in judgement of familiarity and in recency discrimination system, while the hippocampus supports contextual binding (*Eichenbaum et al., 2007*; *Diana et al., 2007*; *Hasselmo, 2005*). These two systems play important roles in memory associative retrieval as suggested by animal studies (*Brown and Aggleton, 2001*), but it is still unclear how the hippocampus and PRC interact during memory processing. *Fell et al. (2006)* suggested that rhinal-hippocampal coupling in the gamma range is associated with successful memory formation. Our results show no evidence for such a phenomenon, but rather agree with more recent studies demonstrating memory-related overall high-frequency desynchronization in the MTL (*Solomon et al., 2019*; *Burke et al., 2015*).

This paper presents the first major application of stochastic volatility models to neural time-series data. The use of a multivariate modeling approach allows us to account for interactions between different subregions in the MTL and thus provides a better fit to the neural data than a univariate approach. Our MSV model fully captures how changes in neural data measured by volatility in one region influences changes in another region, providing insights into the complex dynamics of neural brain signals. We further demonstrated that volatility can be a promising biomarker due to its broadband nature by comparing its performance to one of spectral power in classifying subsequent memory. Finally, researchers can extend these models to broader classes of neural recordings, and exploit their statistical power to substantially increase our understanding of how behavior emerges from the complex interplay of neural activity across many brain regions.

## Materials and methods

### Key resources table

| Reagent, type (species) or resource | Designation | Source or reference | Identifiers | Additional Information |
|---|---|---|---|---|
| Software and algorithm | | *Avants et al., 2008* | http://picsl.upenn.edu/software/ants | advanced normalization tool |
| Software and algorithm | | *Yushkevich et al., 2015* | https://www.nitrc.org/projects/ashs | ashs |

*Continued on next page*

*Continued*

| Reagent, type (species) or resource | Designation | Source or reference | Identifiers | Additional Information |
|---|---|---|---|---|
| Software and algorithm | sklearnc | *Pedregosa et al., 2011* | https://scikit-learn. org/stable/ | |
| Software and algorithm | | This paper | http://memory.psych.upenn. edu/ Electrophysiological_Data | custom processing scripts |
| Software and algorithm | PTSA | This paper | https://github.com/ pennmem/ptsa_new | processing pipeline for reading in iEEG |

## Participants

Ninety six patients with drug-resistant epilepsy undergoing intracranial electroencephalographic monitoring were recruited in this study. Data were collected as part of a study of the effects of electrical stimulation on memory-related brain function at multiple medical centers. Surgery and iEEG monitoring were performed at the following centers: Thomas Jefferson University Hospital (Philadelphia, PA), Mayo Clinic (Rochester, MN), Hospital of the University of Pennsylvania (Philadelphia, PA), Emory University Hospital (Atlanta, GA), University of Texas Southwestern Medical Center (Dallas, TX), Dartmouth-Hitchcock Medical Center (Lebanon, NH), Columbia University Medical Center (New York, NY) and the National Institutes of Health (Bethesda, MD). The research protocol was approved by the Institutional Review Board at each hospital and informed consent was obtained from each participant. Electrophysiological signals were collected from electrodes implanted subdurally on the cortical surface and within brain parenchyma. The neurosurgeons at each clinical site determined the placement of electrodes to best localize epileptogenic regions. Across the clinical sites, the following models of depth and grid electrodes (electrode diameter in parentheses) were used: PMT Depthalon (0.86 mm); Adtech Spencer RD (0.86 mm); Adtech Spencer SD (1.12 mm); Adtech Behnke-Fried (1.28 mm); Adtech subdural and grids (2.3 mm). The dataset can be requested at http://memory. psych.upenn.edu/RAM_Public_Data.

## Free-recall task

Each subject participated in a delayed free-recall task in which they were instructed to study a list of words for later recall test. The task is comprised of three parts: encoding, delay, and retrieval. During encoding, the subjects were presented with a list of 12 words that were randomly selected from a pool of nouns (http://memory.psych.upenn.edu/WordPools). Each word presentation lasts for 1600 ms followed by a blank inter-stimulus interval (ISI) of 800 to 1200 ms. To mitigate the recency effect (recalling last items best) and the primacy effect (recalling first items better than the middle items), subjects were asked to perform a math distraction task immediately after the presentation of the last word. The math problems were of the form A+B+C = ?, where A,B,C were randomly selected digits. The delay math task lasted for 20 s, after which subjects were asked to recall as many words as possible from the recent list of words, in any order during the 30 s recall period. Subjects performed up to 25 lists per session of recording (300 words). Multiple sessions were recorded over the course of the patient's hospital stay.

## Electrophysiological recordings and data processing

iEEG signals were recorded from subdural and depth electrodes at various sampling rates (500, 1000, or 1600 Hz) based on the the amplifier and the preference of the clinical team using one of the following EEG systems: DeltaMed XlTek (Natus), Grass Telefactor, and Nihon-Kohden. We applied a 5 Hz band-stop fourth order Butterworth filter centered on 60 Hz to attenuate signal from electrical noise. We re-referenced the data using the common average of all electrodes in the MTL to eliminate potentially confounding large-scale artifacts and noise. We used Morlet wavelet transform (wave number = 5) to compute power as a function of time for our iEEG signals. The frequencies were sample linearly from 3 to 180 Hz with 1 Hz increments. For each electrode and frequency, spectral power was log-transformed and then averaged over the encoding period. Within a session of recording, the spectral power was z-scored using the distribution of power features across events.

To extract volatility feature, we applied the MSV model to the dataset constructed from the encoding events with an assumption that the parameters governing the dynamics of the volatility process does not change within a session of recording, that is the parameters of the MSV model are assumed to be shared across encoding events. Since we were only interested in the dynamics of the volatility time series of the brain signals, not the orignal time series themselves, we detrended the raw time series using vector autoregressive models of order $p$, where $p$ was selected based on the Akaike information criterion (AIC) to remove any autocorrelation in the raw signals and to make the time series more suited for an MSV application.

In the present manuscript, we used the common average reference (of MTL electrodes) to remove large-scale noise from our MTL recordings. While the bipolar reference is frequently used for such analyses, due to its superior spatial selectivity, several factors limit its utility in this context. First, connectivity between a pair of adjacent bipolar electrodes is contaminated by signal common to their shared monopolar contact; as such, it is difficult to interpret connectivity between such pairs. In the setting of linear depth electrodes placed within the MTL, a substantial portion of the data between pairs of nearby MTL subregions would have to be excluded due to shared monopolar contacts. Second, bipolar re-referencing within the MTL produces the undesirable outcome that a bipolar midpoint 'virtual' electrode could fall in a subregion where neither physical contact was placed, making observed connectivities difficult to interpret.

## Anatomical localization

The MTL electrodes were anatomically localized using the following procedure. Hippocampal subfields and MTL cortices were automatically labeled in a pre-implant 2 mm thick T2-weighted MRI using the Automatic segmentation of hippocampal subfields (ASHS) multi-atlas segmentation method (*Yushkevich et al., 2015*). A post-implant was co-registered with the MRI using Advanced Normalization Tools (*Avants et al., 2008*). MTL depth electrodes that were visible in the CT were then localized by a pair of neuroradiologists with expertise in MTL anatomy.

## Statistical analyses

To assess an effect across subjects, we applied classical statistical tests on the maximum a posteriori (MAP) estimate of the parameter of interest . This approach has been used in many Bayesian applications to FMRI studies (*Stephan et al., 2010*) to test an effect across subjects. For analyses concerning frequencies, we applied Gaussian regression models (*Rasmussen, 2004*) to take the correlations among frequencies into account. We used the Matern (5/2) kernel function for all analyses that used Gaussian regression models, which enforces that the underlying functional form be at least twice differentiable. p-values were FDR-corrected at $\alpha = 0.05$ significance level when multiple tests were conducted.

## Acknowledgements

We thank Blackrock Microsystems for providing neural recording and stimulation equipment. This work was supported by the DARPA Restoring Active Memory (RAM) program (Cooperative Agreement N66001-14-2-4032). We owe a special thanks to the patients and their families for their participation and support of the study. The views, opinions, and/or findings contained in this material are those of the authors and should not be interpreted as representing the official views of the Department of Defense or the U.S. Government. MJK has started a company, Nia Therapeutics, LLC ('Nia'), intended to develop and commercialize brain stimulation therapies for memory restoration and has more than 5% equity interest in Nia. We thank Dr. James Kragel and Nicole Kratz for their thoughtful comments and inputs.

## Additional information

### Funding

| Funder | Grant reference number | Author |
| --- | --- | --- |
| Defense Advanced Research Projects Agency | N66001-14-2-4032 | Michael J Kahana |

The funders had no role in study design, data collection and interpretation, or the decision to submit the work for publication.

### Author contributions
Tung D Phan, Conceptualization, Data curation, Software, Formal analysis, Validation, Visualization, Writing—original draft, Writing—review and editing; Jessica A Wachter, Conceptualization, Supervision, Validation, Writing—review and editing; Ethan A Solomon, Writing—review and editing; Michael J Kahana, Conceptualization, Supervision, Funding acquisition, Investigation, Methodology, Writing—original draft, Writing—review and editing

### Author ORCIDs
Tung D Phan ⓘ https://orcid.org/0000-0001-5957-7566
Ethan A Solomon ⓘ https://orcid.org/0000-0003-0541-7588

### Ethics
Human subjects: Data were collected at the following centers: Thomas Jefferson University Hospital, Mayo Clinic, Hospital of the University of Pennsylvania, Emory University Hospital, University of Texas Southwestern Medical Center, Dartmouth-Hitchcock Medical Center, Columbia University Medical Center, National Institutes of Health, and University of Washington Medical Center and collected and coordinated via the Data Coordinating Center (DCC) at the University of Pennsylvania. The research protocol for the Data Coordinating Center (DCC) was approved by the University of Pennsylvania IRB (protocol 820553) and informed consent was obtained from each participant.

### Decision letter and Author response
Decision letter https://doi.org/10.7554/eLife.42950.025
Author response https://doi.org/10.7554/eLife.42950.026

## Additional files

### Supplementary files
• Transparent reporting form
DOI: https://doi.org/10.7554/eLife.42950.007

### Data availability
The iEEG dataset collected from epileptic patients in this paper is available and, to protect patients' confidentiality, can be requested at http://memory.psych.upenn.edu/RAM_Public_Data. The cmlreaders repository for reading in the data is at https://github.com/pennmem/. The main script for the paper is available at https://github.com/tungphan87/MSV_EEG (copy archived at https://github.com/elifesciences-publications/MSV_EEG).

The following datasets were generated:

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

## Appendix 1

DOI: https://doi.org/10.7554/eLife.42950.008

### MCMC Algorithm for MSV Models

In this section, we derive an MCMC algorithm, which consists of Metropolis-Hastings steps within a Gibbs sampler, for estimating the latent volatility process and its parameters. As before, let $\mathbf{x}_t$ denote the latent multivariate volatility time-series and $\theta = (\mu, \beta, \sum)$ the parameters of the MSV model. Following (*Kim et al., 1998*; *Omori et al., 2007*), we log-transform *Equation 1*

$$y_{j,t}^* = x_{j,t} + \log\left\{ (\epsilon_{j,t}^y)^2 \right\},\qquad (4)$$

where $y_{j,t}^*$ is defined to be $\log(y_{j,t}^2 + c)$ with a fixed offset constant to $c = 10^{-4}$ avoid values equal to 0. *Equation 4* is linear but non-Gaussian. To ameliorate the non-Gaussianity problem, we approximate the log-transformed error term $\log\{(\epsilon_{j,t}^y)^2\} \sim \log(\chi_1^2)$ by a mixture of 10 normal distributions as in *Omori et al. (2007)*:

$$\log(\chi_1^2) \sim \sum_{k=1}^{10} P_k \mathcal{N}(m_k, v_k^2).$$

The values of $p_k, m_k$ and $v_k$ are tabulated in *Omori et al. (2007)*. As a result, we introduce a latent mixture component indicator variable, $r_{j,t}$, for channel $j$ at time $t$ such that $\log\{(\epsilon_{j,t}^y)^2\} \mid (r_{j,t} = k) \sim N(m_k, v_k^2)$. The indicator variable is also estimated in the MCMC sampler. Given the mixture indicator $\mathbf{r}_t$ and the vector parameter $\theta$, the latent volatility series $\mathbf{x}_t$ can be sampled using a forward-filtering and backward-sampling (FFBS) procedure (*West, 1996*). The mixture indicator $\mathbf{r}_t$ can be sampled from a multinomial distribution

$$P(r_{j,t} = k \mid \mathbf{x}_t, \theta) \propto P(r_{j,t} = k)\frac{1}{v_k}\exp\left\{-\frac{(\tilde{y}_{j,t} - x_{j,t} - m_k)^2}{2v_k^2}\right\}.\qquad (5)$$

Finally, the vector parameter $\theta$ can be sampled using an ancillarity-sufficiency interweaving strategy (ASIS) (*Yu and Meng, 2011*; *Kastner and Frühwirth-Schnatter, 2014*) which involves sampling the vector parameter $\theta$ given the unstandardized volatility series $x_{j,t}$ via a Metropolis-Hasting step (non-centered step) and then sampling $\theta$ again given the standardized volatility series $\tilde{x}_{j,t} = \frac{x_{j,t} - \mu_j}{\sigma_j}$ (centered step). *Yu and Meng (2011)* argued that by alternating between the non-centered and centered steps, we obtain a more efficient MCMC sampler that has a better mixing rate and converges faster. In addition, *Kastner and Frühwirth-Schnatter (2014)* showed that the ASIS can accurately sample latent volatility time-series that have low persistences, which is often the case for iEEG signals.

## Appendix 2

DOI: https://doi.org/10.7554/eLife.42950.008

### Parameter Identification

#### Identification with Different Signal-to-noise Ratios

To demonstrate that the Gibbs sampler can accurately estimate the latent volatility process and its associated parameters, we conducted a simulation study in which we generated $N = 5$ time series of length $T = 50,000$ according to *Equations 1 and 2* with various signal-to-noise ratios (SNR), which is controlled by varying the volatility of the volatility series, to mimic the typical length and the number of electrodes in our iEEG datasets. The SNR is calculated by taking the ratio of the average volatility of volatility across electrodes to the expected standard deviation of the noise term in *Equation 4*. We sampled 10,000 posterior draws and discarded the first 5000 draws as a burn-in period to allow for convergence to the stationary distribution. *Appendix 2—table 1* reports the posterior means of the parameters of the MSV model. Throughout the simulation, we use priors whose means are equal to the true values of the parameters. We observe that the Gibbs sampler can reliably estimate the parameters of the MSV model from datasets with various signal-to-noise ratios. The identification of the parameters in the MSV model comes from the strength of our large iEEG dataset which typically has hundreds of thousands of data points per session, an amount of data that rarely exists in any financial application.

**Appendix 2—table 1.** Parameter Recovery.

| Dataset | SNR | Channel | Truth | | | | | | | MSV | | | | | | |
|---|---|---|---|---|---|---|---|---|---|---|---|---|---|---|---|---|
| | | | $\alpha$ | $b_1$ | $b_2$ | $b_3$ | $b_4$ | $b_5$ | $\sigma$ | $\alpha$ | $b_1$ | $b_2$ | $b_3$ | $b_4$ | $b_5$ | $\sigma$ |
| 1 | 0.16 | 1 | 3.39 | 0.85 | 0.20 | 0.00 | 0.00 | 0.00 | 0.17 | 3.39 | 0.86 | 0.19 | 0.00 | -0.04 | 0.00 | 0.17 |
| | | 2 | 3.60 | 0.00 | 0.88 | -0.10 | 0.00 | 0.00 | 0.19 | 3.60 | 0.00 | 0.87 | -0.10 | -0.00 | -0.01 | 0.19 |
| | | 3 | 3.55 | 0.00 | 0.00 | 0.87 | 0.30 | 0.00 | 0.19 | 3.53 | -0.00 | -0.00 | 0.85 | 0.29 | 0.01 | 0.20 |
| | | 4 | 3.51 | 0.00 | 0.00 | 0.00 | 0.71 | 0.00 | 0.12 | 3.51 | 0.01 | -0.00 | 0.01 | 0.69 | -0.03 | 0.14 |
| | | 5 | 3.38 | 0.00 | 0.00 | 0.00 | 0.00 | 0.80 | 0.15 | 3.38 | 0.00 | -0.01 | -0.01 | 0.06 | 0.79 | 0.15 |
| 2 | 0.27 | 1 | 3.60 | 0.95 | 0.20 | 0.00 | 0.00 | 0.00 | 0.21 | 3.93 | 0.95 | 0.19 | -0.00 | 0.00 | 0.00 | 0.24 |
| | | 2 | 3.74 | 0.00 | 0.90 | -0.10 | 0.00 | 0.00 | 0.25 | 3.82 | -0.00 | 0.90 | -0.10 | 0.00 | -0.00 | 0.25 |
| | | 3 | 3.89 | 0.00 | 0.00 | 0.93 | 0.30 | 0.00 | 0.29 | 3.81 | -0.00 | 0.00 | 0.93 | 0.30 | 0.00 | 0.29 |
| | | 4 | 3.05 | 0.00 | 0.00 | 0.00 | 0.91 | 0.00 | 0.28 | 3.04 | -0.00 | 0.01 | 0.01 | 0.91 | 0.00 | 0.28 |
| | | 5 | 3.96 | 0.00 | 0.00 | 0.00 | 0.00 | 0.95 | 0.22 | 3.97 | -0.00 | 0.00 | 0.00 | -0.00 | 0.94 | 0.22 |
| 3 | 0.42 | 1 | 3.25 | 0.52 | 0.20 | 0.00 | 0.00 | 0.00 | 0.42 | 3.25 | 0.51 | 0.20 | 0.01 | -0.02 | 0.01 | 0.43 |
| | | 2 | 3.64 | 0.00 | 0.57 | -0.10 | 0.00 | 0.00 | 0.46 | 3.63 | 0.01 | 0.55 | -0.14 | 0.02 | 0.02 | 0.46 |
| | | 3 | 3.18 | 0.00 | 0.00 | 0.58 | 0.30 | 0.00 | 0.31 | 3.18 | 0.03 | 0.01 | 0.56 | 0.32 | 0.03 | 0.31 |
| | | 4 | 3.44 | 0.00 | 0.00 | 0.00 | 0.65 | 0.00 | 0.41 | 3.45 | 0.00 | -0.00 | -0.00 | 0.64 | 0.00 | 0.41 |
| | | 5 | 3.26 | 0.00 | 0.00 | 0.00 | 0.00 | 0.60 | 0.36 | 3.26 | -0.02 | -0.00 | 0.00 | -0.01 | 0.60 | 0.36 |

We generated three datasets with different signal-to-noise ratios. The observed multivariate time-series $y_t$ was simulated according to the data-generating process specified by the MSV model with pre-specified parameters (truth). We then applied the MSV model to the simulated series $y_t$ to recover the parameters of the MSV model. In this simulation study, the non-zero off-diagonal entries of the matrix $\beta$ were fixed across datasets. The diagonal elements of $\beta$ were generated from a uniform distribution on [0.7, 0.9], [0.9, 1.0], and [0.5, 0.7] respectively. The volatilities of volatility of the electrodes were generated from a uniform distribution on [0.1, 0.2], [0.2, 0.3], and [0.3, 0.5] respectively.

DOI: https://doi.org/10.7554/eLife.42950.010

## Appendix 3

DOI: https://doi.org/10.7554/eLife.42950.008

### How Much Data is Enough?

Here we provide an analysis of the amount of data needed to estimate the MSV model. The number of parameters of the MSV model is on the order of $O(J^2)$, where $J$ is the dimension of the multivariate time series. In our case, $J$ is just the number of recording locations. To justify this claim, we observe in **Equation 2** that the variable reflecting the unconditional mean $\mu$ has $J$ parameters, the persistent matrix $\beta$ has $J^2$ parameters and the covariance matrix $\Sigma$ has $J$ parameters, totaling to $J + J^2 + J = J^2 + 2J = O(J^2)$ parameters. As the dimension of the timeseries increases, the amount of data required will need to increase quadratically for a good estimation of the model.

To assess the performance of various time lengths, we repeated the simulation study in Appendix .2 in which we simulated 30 datasets for each $T = k(J^2 + 2J)$ for k ranging from 30 to 180 with an increment of 30. We used the log-determinant error, which is defined to be

$$\log - \mathrm{determinant\, error} = \log(|\det(\beta_{MSV}^{-1}\beta_{true} - I_J)|), \tag{6}$$

where $\beta_{MSV}$ is the estimated connectivity matrix from the MSV model and $\beta_{true}$ is the true simulated connectivity matrix. If $\beta_{MSV}$ is close to $\beta_{true}$, then the $\beta_{MSV}^{-1}\beta$ is close to the identity matrix $I_J$, and therefore, the log determinant error will be very negative.

**Appendix 3—figure 1** shows that as we increase the amount of data, the log-determinant error becomes more negative. With $T = 30(J^2 + 2J)$, the average determinant error is approximately $8.5 \times 10^{-9}$, which is very close . Therefore, we recommend the amount of data to be at least $30O(J^2)$ for the MSV model.

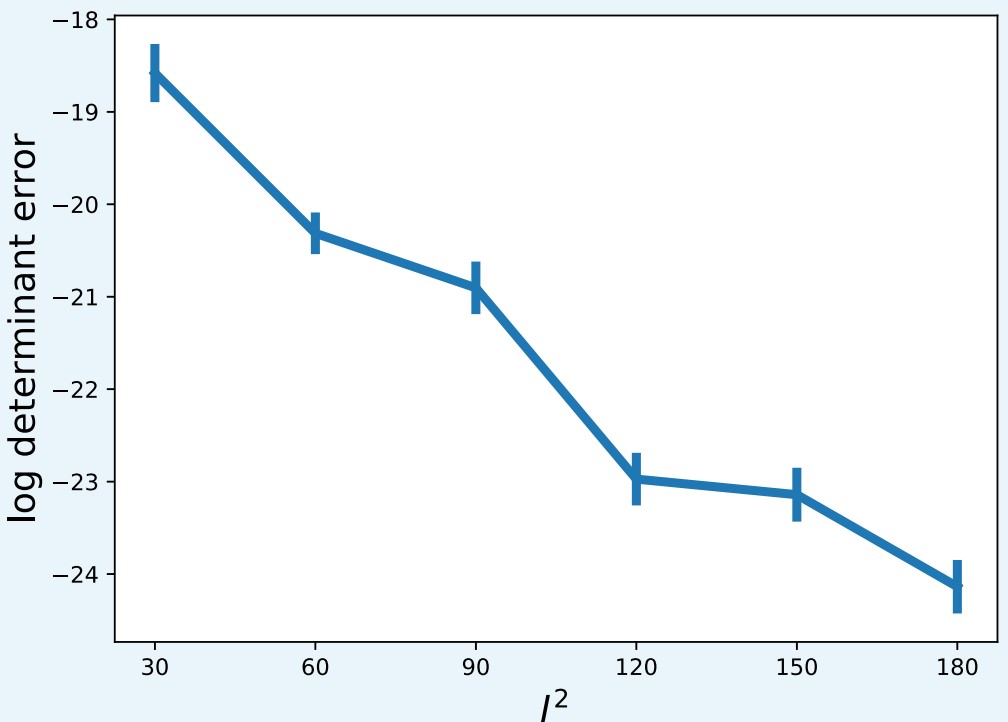

**Appendix 3—figure 1.** Time length analysis. For $k$ ranging from 30 to 180 with 30 increments and for each $T = k(J^2 + 2J)$, we simulated 30 datasets according to the data-generating process specified by the MSV model. Then, we estimated the connectivity matrix $\beta_{MSV}$ for each

dataset. We assess the performance of the MSV model in estimating the true connectivity matrix $\beta_{true}$ using the log-determinant error metric, $\log(|\det(\beta_{MSV}^{-1}\beta_{true} - I_J)|)$. The figure shows the average performance at each time length with the standard error bars.
DOI: https://doi.org/10.7554/eLife.42950.012

Appendix 3—figure 2 demonstrates that the MSV model can recover the latent volatility timeseries with $T = 30(J^2 + 2J)$, validating the algorithm that is used to estimate the paramters of the model.

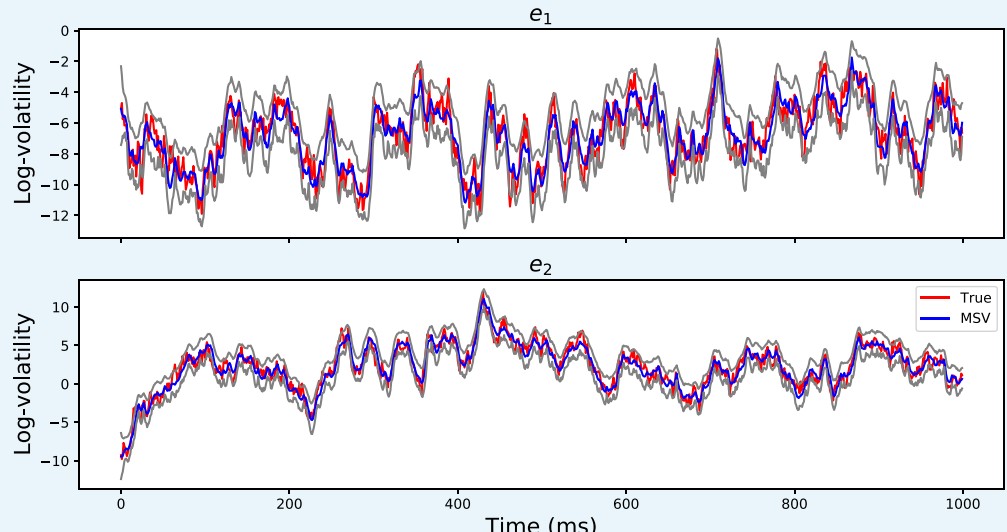

**Appendix 3—figure 2.** Volatility Timeseries Recovery. The red lines show the true simulated log-volatility timeseries for the first 1000 time points for two channels . The blue timeseries show the estimated log-volatility time series using the MSV model and the gray timeseries are the 95% posterior confidence intervals. The figure demonstrates that the MSV model can estimate the latent log-volatility timeseries well.
DOI: https://doi.org/10.7554/eLife.42950.013

# Appendix 4

DOI: https://doi.org/10.7554/eLife.42950.008

## Model Fit Plots

We provide a visualization of the latent volatility series. *Appendix 4—figure 1* illustrates the recordings from a hippocampal electrode during encoding of a list of 12 word items from a subject performing a verbal free-recall task at the University of Pennsylvania Hospital. The top panels show the detrended iEEG series using AR(p) models. The bottom panels show the respective latent volatility series associated with the detrended signals. We observe that the latent volatility series capture the instantaneous variance of the original series.

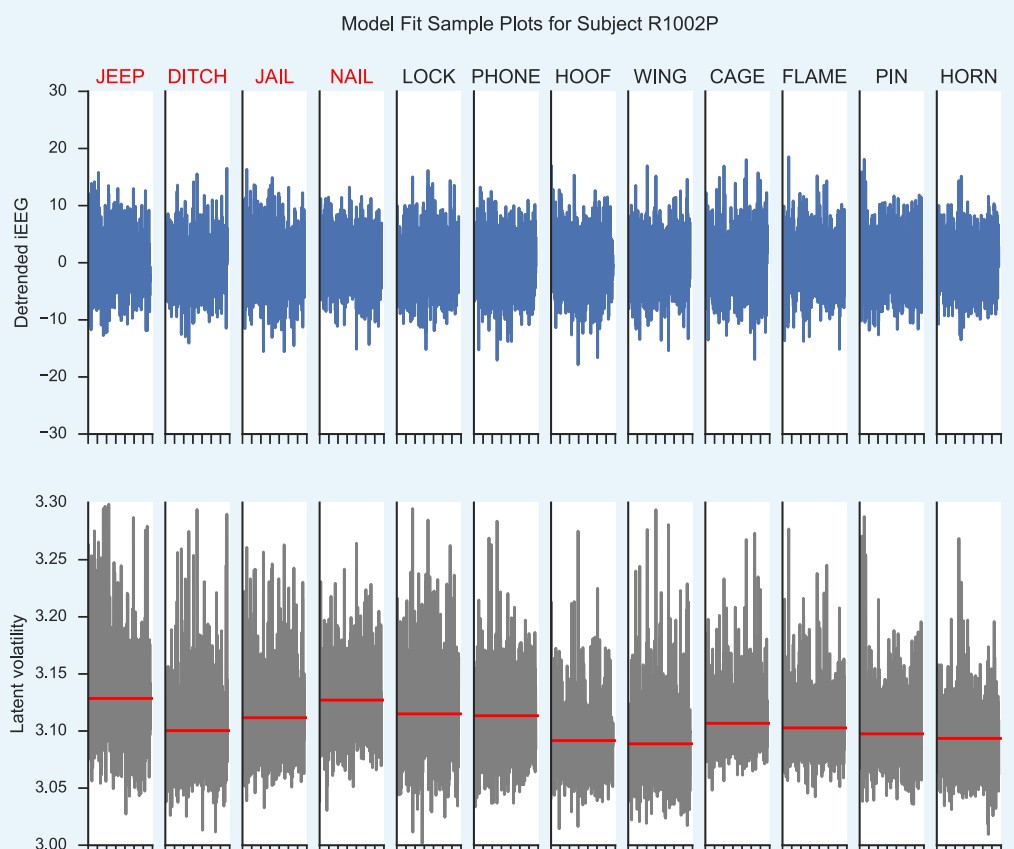

**Appendix 4—figure 1.** Model fit plots for a hippocampal electrode. The upper panels show the detrended iEEG signals using an AR(p) model for encoding periods of a list of words. The lower panels show the associated estimated latent volatility processes. The red lines indicate the average volatility during the encoding period. Red words are later recalled and blue words are not recalled.

DOI: https://doi.org/10.7554/eLife.42950.015

# Appendix 5

DOI: https://doi.org/10.7554/eLife.42950.008

## Model Selection Details

### Deviance Information Criterion (DIC)

The deviance information criterion introduced in **Spiegelhalter et al. (2002)** is a Bayesian analogue of the Akaike information criterion (AIC), which is typically used to assess model performance. Given the observed data y and a parameter set $\theta$ of a model, the AIC is defined to be

$$\text{AIC} = \underbrace{-2\log\{p(\mathbf{y} \mid \hat{\theta})\}}_{\text{goodnessoffit}} + \underbrace{2k}_{\text{numberofparameters}} , \tag{7}$$

where $\hat{\theta}$ is the maximum likelihood estimate and $k$ is the number of parameters of the model. The AIC is just the negative log-likelihood (deviance) $D(\theta) = \log\{p(\mathbf{y} \mid \hat{\theta})\}$ penalized by the number of parameters used in the model. AIC has been shown to be asymptotically equivalent to leave-one-out cross-validation (**Stone, 1977**). A small AIC indicates that the model fits the data well after accounting for the number of parameters. Under the MCMC framework, **Spiegelhalter et al. (2002)** proposed a Bayesian equivalence of **Equation 7** such that the maximum likelihood estimate $\hat{\theta}$ is replaced by the posterior mean of $\tilde{\theta} = E[\theta \mid \mathbf{y}]$ and the number of parameters is replaced by the effective number of parameters $p_D = E_{\theta \mid \mathbf{y}}[-2\log p(\mathbf{y} \mid \theta)] + 2\log\{p(\mathbf{y} \mid \tilde{\theta})\}$

$$\text{DIC} = \underbrace{-2\log\{p(\mathbf{y} \mid \tilde{\theta})\}}_{\text{goodnessoffit}} + \underbrace{2p_D}_{\text{effectivenumberofparameters}} . \tag{8}$$

The calculation of DIC can be readily done using the samples from the posterior distribution of $P(\theta \mid \mathbf{y})$. Next we will derive the calculation of DIC for the MSV model.

### DIC Calculation for MSV Model

The MCMC algorithm derived in Appendix 1 provides a way to sample from the posterior distribution of $\theta = (\mathbf{x}, \mu, \beta, \Sigma) \mid \mathbf{y}$. We assume that there are $K$ (typically 1000) samples $\{\theta^{(1)}, \cdots \theta^{(K)}\}$ from the posterior distribution $P(\theta \mid \mathbf{y})$. The posterior mean in Eqn. in eight can be estimated by $\tilde{\theta} = (\tilde{\mathbf{x}}, \tilde{\mu}, \tilde{\beta}, \tilde{\Sigma}) = \frac{1}{K}\sum_{k=1}^{K}\theta^{(k)}$. The goodness-of-fit term can be calculated as follows:

$$-2\log p(\mathbf{y} \mid \tilde{\theta}) = -2\log p(\mathbf{y} \mid \tilde{\mathbf{x}}, \tilde{\mu}, \tilde{\beta}, \tilde{\Sigma}) \tag{9}$$

$$= -2\sum_{t=1}^{T}\log p(y_t \mid \tilde{x}_t) \tag{10}$$

$$= -2\sum_{t=1}^{T}\log \phi(y_t \exp(-\tilde{x}_t/2)), \tag{11}$$

where $\phi$ is the probability density function of the standard normal distribution. To calculate the effective number of parameters, we first need to compute the posterior mean of the log-likelihood function. This can be done using Monte Carlo integration

$$E_{\theta \,|\, \mathbf{y}}[-2 \log p(\mathbf{y} \mid \theta)] = \frac{-2}{K} \sum_{k=1}^{K} \log p(\mathbf{y} \mid \theta^{(k)}) \tag{12}$$

$$= \frac{-2}{K} \sum_{k=1}^{K} \sum_{t=1}^{T} \log \phi(y_t \exp(-x_t^{(k)}/2)). \tag{13}$$

## Appendix 6

DOI: https://doi.org/10.7554/eLife.42950.008

### Classification of Subsequent Memory Details

In this section, we explain the details of the penalized logistic regression classifier and the nested cross-validation approach (**Krstajic et al., 2014**) that is used to evaluate the classifier's predictive performance. We ask whether the implied volatility features across brain regions during the encoding period can reliably predict subsequent memory recall. Let $\{y_i\}_{i=1}^N$ be the binary outcomes (recalled versus non-recalled) of $N$ word-trials and $\{\mathbf{X}_i\}_{i=1}^N$ be their corresponding features of interest (volatility or spectral power). Here, we wish to distinguish $y_i$ and $\mathbf{X}_i$, which are the output and input variables of the classification model, from $\{\mathbf{y}_t\}_{t=1}^T$ and $\{\mathbf{X}_t\}_{t=1}^T$, which are iEEG and volatility timeseries respectively. The penalized L-2 logistic classifier aims to find the optimal set of weights $\omega$ that minimizes the cross-entropy between the outcome variable $y_i$ and the logistic transformation of $\omega^T \mathbf{X}_i$ while limiting the contributions of individual features by penalizing the L2-norm of $\omega$

$$l(\omega) = \underbrace{-\lambda \sum_{i=1}^N y_i \log \hat{y}_i + (1 - y_i) \log(1 - \hat{y}_i)}_{cross-entropy} + \underbrace{0.5 \|\omega\|_2^2}_{penalty}, \tag{14}$$

where $\hat{y}_i = \dfrac{\exp(\omega^T \mathbf{X}_i)}{1 + \exp(\omega^T \mathbf{X}_i)}$ is the predicted probability of recall success, and $\lambda$ is the penalty parameter that controls the degree to which the classifier limits the contribution of individual features. A low value of $\lambda$ indicates that the classifier downgrades the contribution of the cross-entropy in the loss function, and therefore penalizing the L-2 norm of $\omega$ more severely and vice versa. In addition, minimizing the cross-entropy loss between the empirical distribution (observed data) and distribution given by the model (predicted data) is equivalent to maximizing the log-likelihood function (**Goodfellow et al., 2016**). To test the ability of the classifier to generalize to new data, we employ a k-fold nested cross-validation approach in which we split the data naturally into different recording sessions (folds) illustrated in **Appendix 6—figure 1**. We hold out one session as the test data and train the classifier on the remaining session(s). The optimal hyperparameter $\lambda$ is optimized via another cross-validation within the training set. The procedure is repeated until we obtain out-of-sample prediction for the entire dataset.

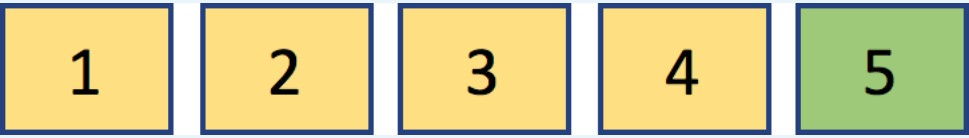

**Appendix 6—figure 1.** Cross-validation scheme. For each subject, we train our classifier on all (yellow) but one session and test the performance on the hold-out session (green) and repeat the procedure for each session.

DOI: https://doi.org/10.7554/eLife.42950.018

We assess the classifier's performance using the area under the receiver operating curve (AUC), which relates the true positive rate and the false positive rate for various decision thresholds. An AUC of 1.0 implies that the classifier can perfectly discriminate between recalled and non-recalled items and an AUC of 0.5 implies that the classifier is at chance. **Appendix 6—figure 2** illustrates the classifier's predictive performance for two subjects. The ROCs for these subjects sit above the 45-degree line, indicating that the classifier can reliably discriminate subsequent memory recall above chance.

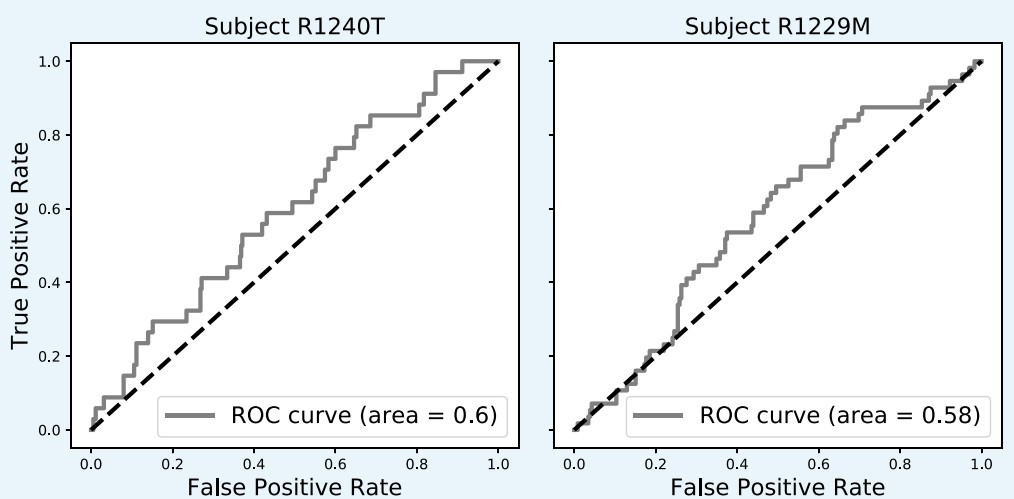

**Appendix 6—figure 2.** Sample ROC curves (gray). The black dashed lines indicate ROC curves of at-chance classifiers. The area under the ROC curve (AUC) is a measure of the performance of the classifier across a spectrum of decision thresholds. An AUC of 1 indicates that the classifier can classify recalled items perfectly. An AUC of 0.5 indicates that the classifier is at chance, that is as good as a random coin flip.

DOI: https://doi.org/10.7554/eLife.42950.019

## Appendix 7

DOI: https://doi.org/10.7554/eLife.42950.008

### Output Tables of Statistical Tests

This section reports the statistical tests for directional connection SME that contain at least 10 subjects. *Appendix 7—tables 1* and *2* show the results of these tests for the left and right hemispheres.

**Appendix 7—table 1.** intra-MTL directional connectivity in the left hemisphere.

| Region I → region J | Mean $\Delta_{I \to J}$ | Se | T | N | P | adj. P |
|---|---|---|---|---|---|---|
| Hipp → PRC | −0.044 | 0.013 | −3.494 | 48 | 0.001 | 0.009** |
| PRC → Hipp | −0.060 | 0.022 | −2.667 | 48 | 0.011 | 0.045* |
| Hipp → EC | 0.010 | 0.032 | 0.312 | 14 | 0.768 | 0.953 |
| EC → Hipp | −0.077 | 0.067 | −1.158 | 14 | 0.285 | 0.569 |
| PHC → PRC | −0.006 | 0.040 | −0.146 | 16 | 0.889 | 0.953 |
| PRC → PHC | −0.037 | 0.028 | −1.348 | 16 | 0.212 | 0.564 |
| PRC → EC | 0.001 | 0.022 | 0.061 | 21 | 0.953 | 0.953 |
| EC → PRC | 0.005 | 0.030 | 0.168 | 21 | 0.872 | 0.953 |

DOI: https://doi.org/10.7554/eLife.42950.021

**Appendix 7—table 2.** intra-MTL directional connectivity in the right hemisphere.

| Region I → region J | Mean $\Delta_{I \to J}$ | Se | T | N | P | adj. P |
|---|---|---|---|---|---|---|
| Hipp → PRC | −0.010 | 0.020 | −0.471 | 40 | 0.645 | 0.838 |
| PRC → Hipp | −0.016 | 0.029 | −0.575 | 40 | 0.574 | 0.838 |
| Hipp → EC | 0.011 | 0.030 | 0.361 | 14 | 0.733 | 0.838 |
| EC → Hipp | 0.054 | 0.047 | 1.144 | 14 | 0.290 | 0.838 |
| Hipp → PHC | −0.027 | 0.032 | −0.837 | 15 | 0.432 | 0.838 |
| PHC → Hipp | 0.044 | 0.035 | 1.232 | 15 | 0.254 | 0.838 |
| PRC → EC | 0.020 | 0.052 | 0.378 | 14 | 0.722 | 0.838 |
| EC → PRC | −0.002 | 0.081 | −0.020 | 14 | 0.985 | 0.985 |

∗ : P<0.05, ∗∗ : P<0.01, adjusted *p*-values were calculated using the Benjamini-Hochberg procedure.

DOI: https://doi.org/10.7554/eLife.42950.022

