## [Decision Letter]

Thank you for submitting your article "Multivariate Stochastic Volatility Modeling of Neural Data" for consideration by *eLife*. Your article has been reviewed by three peer reviewers, and the evaluation has been overseen by a Reviewing Editor and Michael Frank as the Senior Editor. The following individual involved in review of your submission has agreed to reveal their identity: Pedro Valdés-Sosa (Reviewer #3).

The reviewers have discussed the reviews with one another and the Reviewing Editor has drafted this decision to help you prepare a revised submission.

Summary:

The authors employ time varying analyses of the (multivariate) variance to a large data set of intracranial EEG. A hierarchical series of analyses is applied to capture an increasing complexity of distributed network activity. The authors establish that their approach can capture neuronal activity that predicts the behavioural performance of the experimental participants, an important proof-of-principle.

This is an important departure from classic (mean-based) autoregressive models of neurophysiological data. All three reviewers acknowledge the innovation of the approach and the quality of the presentation.

Essential revisions:

1) Better contextualization is requested by all three reviewers.

2) Reviewer 1 requests some further validation and I agree with this. You may choose to use *eLife*'s figure supplements to do this.

3) Reviewer 1 also requests some elaboration of the theory and better presentation of the figures. These should be addressed.

4) I think the comparison with the power spectrum in the prediction section is informative in its current state (the authors already attended to aspects of this section in a pre-submission process). Reviewer 1 has requested consideration of some alternative benchmarks here – these may be of interest but are not essential.

*Reviewer #1:*

The manuscript “Multivariate Stochastic Volatility Modeling of Neural Data” by Phan et al. addresses the problem of time-varying variance (volatility) in EEG data. It presents a multivariate model that describes the evolution of the variance over time. In this multivariate model, the variances of the channels evolve in a fashion described by an autoregressive model. The authors present a method to fit the model to data. They then (i) show that the multivariate model captures the data better than a univariate model, (ii) show that the volatility approach correlates with established methods based on the power spectrum, (iii) demonstrate that the volatility bears predictive power for the outcome of the experiment, and (iv) use the volatility coupling matrix to infer some changes in the effective connectivity between successfully memorized and not memorized items.

While their approach is intriguing in places, and the inference of multivariate couplings is indeed an open topic in the field, I have several considerations that limit my enthusiasm. I would request the authors to address these in detail before reconsidering the manuscript.

First, the authors introduce a novel method for estimating model parameters without showing many of its central properties. Most importantly, the data-sensitivity of multivariate approaches is not explored here (although the authors do acknowledge it in the Introduction): how does the variability of their method depend on the amount of data? Appendix 1—table 1 does not address these problems. Furthermore, the three example studies are not a systematic proof of the consistency of the estimate. What happens, for example, if the largest eigenvalue of the coupling matrix is close to 1, i.e. close to a phase transition?

Second, in this light the authors should at least compare their method to established pair-wise quantities (cross-correlations, synchronization, information theoretic measures) and show that the observed decoupling is not equally visible in these well-established (and more data-efficient) methods.

Third, there is considerable literature on multivariate time series analysis, including volatility. For example, ARCH and GARCH models are rather frequently used. One reference that I am aware of (although this is just the tip of the iceberg) is Galka et al. (2010). The authors should relate their work to the established models, show how they differ and what benefit their novel approach yields.

Equation 1: I do not really understand why the model is constructed this way. I understand that the modeled variance enters exponentially, but do the authors have a justification for this approach? Second, the approach presented here only correlates the variance over time, while ϵjty is still drawn from a normal distribution. If I am not mistaken, *y_t_* will hence look like Gaussian noise with changing amplitude. Does the data look like this? I would expect a more pronounced autocorrelation of the raw EEG signals. It might help if the authors showed (and clearly marked, see below about Figure 1) example data that they try to fit.

Figure 1: I find panel C inconsistent and incomprehensible in many respects. (i) Why are there 1 second intervals shown in the Countdown period? Why is this panel shown at all, given there are no data? (I guess this indicates that these data were not analyzed?) (ii) In the encoding period, why are there pauses between the words? These pauses are not mentioned in the description of the experiment. Are they included in the analysis or not? If not, why are the data still shown, given no data are shown for the Countdown period? Also, the label indicating the length of the intervals is rather hidden, given all other scales are at the top of the diagrams. Last, what does REM / Not REM mean? It can be guessed but is not explained anywhere. (iii) A distractor task following the encoding is mentioned in the text but not shown in the figure. (iv) The free-recall diagram is not explained at all. Was it analyzed (so far, the text says otherwise)? If not, why are data still shown, and what do the intervals indicate? Also there is no length scale for the white intervals. (v) The panel does not state what the plots show. The (very small, and unintuitively placed) label "Compute volatility" seems to indicate that it is the time series *y_t,i_* of two channels, from which the volatility is then computed? (vi) I think readers could greatly benefit from an illustration of the workflow from y to theta and x, for example incorporating elements from Appendix 1—figure 1.

Subsection “Relation to Spectral Power”: This whole paragraph is imprecise in the details. (i) What is an "encoding event"? If it is each single word during any encoding period, this would imply a data length of 1.6 seconds each. How would this fit with the Introduction, which states that fitting a multivariate model is hard for limited data? (ii) The Materials and methods say that averaging was done per subject. So does Figure 2 show exemplary results for one subject? If so, this should be indicated, and the generalization to all subjects discussed.

Subsection “Classification of Subsequent Memory Recall”: I get that the authors want to show the predictive power of volatility. However, to make this point it would suffice to show that it is above chance level. I do not see the significance of comparing it to the predictive power of methods based on the power spectrum. Furthermore, why does the classifier only take into account each band of the power spectrum? Should a classification based on all bands simultaneously not be much more powerful?

Subsection “Directional Connectivity Analysis”: While I understand the approach by the authors, the results are displayed in a way that I find hard to comprehend. (i) The authors should indicate the inferred values for the coupling matrix and discuss its significance for the dynamics of the process (intrinsic timescales etc.). (ii) The authors first state that directed connections between left HC and PRC decrease, but then also say that the difference between directional connections between these areas is not significant. Do they refer to the difference between HC → PRC and PRC → HC? If yes, do the reported p-values refer to the R or NR set of items?

The first three paragraphs of the Discussion are a mere introduction or repetition of results, and should hence be included in the Introduction or be omitted.

Discussion, fourth paragraph: Why is the decoupling only found in the left hemisphere? This striking difference is completely neglected by the authors. Although I am not too familiar with the cited study by Fell et al. (2006), it does not appear to confirm this difference between hemispheres. As this limitation to the left hemisphere will probably be very obvious to many readers, the authors should discuss it.

Discussion, last paragraph: The authors suggest that fellow researchers can use and extend their method. With this in mind, it would be very beneficial if the authors made their code publicly available.

*Reviewer #2:*

I believe that this paper succeeds in rolling out a new method for multivariate signal analysis for multi-channel neural data. The paper presents an application of an approach to multi-process (here multi-channel) time series data that was first utilized in the analysis of financial time series. In this approach, the variance of the time series is not assumed to be constant across the periods of analysis. Methods such as Granger causality assume variance stationarity between brain regions in order to identify directed interactions between the regions. However, variance stationarity is often not observed in neural data. The method developed here (MSV: multivariate stochastic volatility) models multi-electrode depth recordings from the medial temporal lobes (MTL) of 96 epilepsy patients participating in word-list recall task. The key innovation is the use of an autoregressive framework to model the fluctuations in the variance of the time series across the encoding interval of the behavioral task and across the data channels. Through a fairly rigorous and detailed analysis of the variables of the state-space MSV model, including the regression coefficient matrix, the authors are able to infer how the signals in the multiple channels interact in time and across space. The authors are able to conclude that during successful encoding of words the connectivity between the hippocampus and the perirhinal cortex in the left hemisphere decreases from a resting condition where the former drives the latter more strongly than the latter drives the former. The authors are also able to predict which words from the lists will be successfully recalled by measuring the volatility in the neural data.

The methodology described here will find many applications in other studies using large-scale electrophysiology and MEG. As a consequence, this paper will be of considerable interest to the large community of researchers investigating brain networks.

The authors use a limited number of equations to explain their methods of modeling and their analysis of the data and instead rely on the text do a lot of the expository work, which is fine given that text is clear and well written.

The authors do a good job of placing the MSV model in the context of non-parametric methods such as spectral analysis and provide a means to link the two together. Through this linkage the authors are able to show that the correlation between volatility and spectral power is nearly proportional to the analysis frequency in the spectrum. This suggests that neural activity in the gamma band is more volatile than at lower frequencies. This result is of interest because it suggests that the conflicting reports in the literature about the nature and function of gamma band activity in neural activity may in part be due to the application of non-parametric methods that require that the time series be stationary over the data periods of interest.

One improvement here would be if the authors also included the analysis of the neural activity during a time window other than the encoding phase. For example, if the countdown phase was also submitted to the same MSV modeling, the hippocampus and perirhinal cortex should show a different functional connectivity than what was reported for successful encoding of words. The authors should also comment on whether the MSV modeling can be applied to a network that includes 20 distinct sites, instead of just 4, as was done in the paper.

*Reviewer #3:*

This is excellent work and directs us away from looking at directional measure of influence based on pure autoregressive models (which focus on the conditional mean). Rather interest is focused on the conditional variance, which is shown in a clear way to have advantages over traditional Granger Causality measures. I am convinced by the evidence presented.

The paper would benefit by clarifying the sub-types of stochastic volatility models. There is some confusion in the field. The authors seem to be applying a type of GARCH model, sometimes proposed as distinct from SV and sometimes a subtype.

I do suggest that some prior work be cited properly. The Wong paper is cited in a way that suggests that it is relevant for emphasizing non-stationary when in reality is also a multivariate stochastic volatility model. The same group even attempted source localization in Galka, Yamashita and Ozaki (2004). This is all well covered in the book by Ozaki on time series modeling of neuroscience data.

Finally in the Discussion I would suggest a discussion with combined AR type models and those with stochastic volatility: see Mohamadi et al. (2017).

I also do not see if the code will be made publicly available.

---

## [Author Response]

Reviewer #1:[…] While their approach is intriguing in places, and the inference of multivariate couplings is indeed an open topic in the field, I have several considerations that limit my enthusiasm. I would request the authors to address these in detail before reconsidering the manuscript.First, the authors introduce a novel method for estimating model parameters without showing many of its central properties. Most importantly, the data-sensitivity of multivariate approaches is not explored here (although the authors do acknowledge it in the Introduction): how does the variability of their method depend on the amount of data? Appendix 1—table 1 does not address these problems. Furthermore, the three example studies are not a systematic proof of the consistency of the estimate. What happens, for example, if the largest eigenvalue of the coupling matrix is close to 1, i.e. close to a phase transition?Second, in this light the authors should at least compare their method to established pair-wise quantities (cross-correlations, synchronization, information theoretic measures) and show that the observed decoupling is not equally visible in these well-established (and more data-efficient) methods.Third, there is considerable literature on multivariate time series analysis, including volatility. For example, ARCH and GARCH models are rather frequently used. One reference that I am aware of (although this is just the tip of the iceberg) is Galka et al. (2010). The authors should relate their work to the established models, show how they differ and what benefit their novel approach yields.

*Equation 1: I do not really understand why the model is constructed this way. I understand that the modeled variance enters exponentially, but do the authors have a justification for this approach? Second, the approach presented here only correlates the variance over time, while* ϵjty *is still drawn from a normal distribution. If I am not mistaken, y_t_ will hence look like Gaussian noise with changing amplitude. Does the data look like this? I would expect a more pronounced autocorrelation of the raw EEG signals. It might help if the authors showed (and clearly marked, see below about Figure 1) example data that they try to fit.*

We apologize for not providing enough details on the construction of the MSV model. The exponential term is actually the variance process, which is always positive. We model the log-variance using a vector autoregressive process since the variance is approximately log-normally distributed as demonstrated in the “Volatility of IEEG is Stochastic” section. In addition, we work with the detrended timeseries (after removing autoregressive components) instead of the original voltage series. The new Figure 1 demonstrates the process of going from the raw iEEG data to the volatility timeseries.

Figure 1: I find panel C inconsistent and incomprehensible in many respects. (i) Why are there 1 second intervals shown in the Countdown period? Why is this panel shown at all, given there are no data? (I guess this indicates that these data were not analyzed?). (ii) In the encoding period, why are there pauses between the words? These pauses are not mentioned in the description of the experiment. Are they included in the analysis or not? If not, why are the data still shown, given no data are shown for the Countdown period? Also, the label indicating the length of the intervals is rather hidden, given all other scales are at the top of the diagrams. Last, what does REM / Not REM mean? It can be guessed but is not explained anywhere. (iii) A distractor task following the encoding is mentioned in the text but not shown in the figure. (iv) The free-recall diagram is not explained at all. Was it analyzed (so far, the text says otherwise)? If not, why are data still shown, and what do the intervals indicate? Also there is no length scale for the white intervals. (v) The panel does not state what the plots show. The (very small, and unintuitively placed) label "Compute volatility" seems to indicate that it is the time series y_t,i_ of two channels, from which the volatility is then computed? (vi) I think readers could greatly benefit from an illustration of the workflow from y to theta and x, for example incorporating elements from Appendix 1—figure 1.

We replace Figure 1 in the previous manuscript with Figure 2 in the current manuscript. Figure 2 now includes the three relevant phases of the free-recall task, the recording locations, and the work flow of the MSV model.

Subsection “Relation to Spectral Power”: This whole paragraph is imprecise in the details. (i) What is an "encoding event"? If it is each single word during any encoding period, this would imply a data length of 1.6 seconds each. How would this fit with the Introduction, which states that fitting a multivariate model is hard for limited data? (ii) The Materials and methods say that averaging was done per subject. So does Figure 2 show exemplary results for one subject? If so, this should be indicated, and the generalization to all subjects discussed.

We apologize for not being clear. An encoding event is just a single word event. We apply the MSV model to the combined dataset of all encoding events with the assumption that these events share the same set of parameters which govern their volatility processes. This assumption is also stated in the Materials and methods section. Figure 2 in the previous manuscript now becomes Figure 3. In Figure 3, we clearly state that the confidence bands are constructed across subjects. Since each subject has multiple sessions of recordings, we run a separate MSV model for each session within a subject and the results are averaged across sessions.

Subsection “Classification of Subsequent Memory Recall”: I get that the authors want to show the predictive power of volatility. However, to make this point it would suffice to show that it is above chance level. I do not see the significance of comparing it to the predictive power of methods based on the power spectrum. Furthermore, why does the classifier only take into account each band of the power spectrum? Should a classification based on all bands simultaneously not be much more powerful?

We compare the performance of volatility to that of spectral power at a specific frequency in predicting subsequent memory recall to ensure that the two classifiers have the same number of features, thus making it a fair comparison. As you suggested, the classifier based on all bands yields a higher performance due to having a lot more information. What our manuscript suggests is that volatility contains more information than each of the individual frequencies.

*Subsection “Directional Connectivity Analysis”: While I understand the approach by the authors, the results are displayed in a way that I find hard to comprehend. (i) The authors should indicate the inferred values for the coupling matrix and discuss its significance for the dynamics of the process (intrinsic timescales etc.). (ii) The authors first state that directed connections between left HC and PRC decrease, but then also say that the difference between directional connections between these areas is not significant. Do they refer to the difference between HC → PRC and PRC* → *HC? If yes, do the reported p-values refer to the R or NR set of items?*

(i) The inferred values reflect the lag-one correlations among different locations in the brain, which are explained in the construction of the MSV model. (ii) Yes, the reviewer is correct. We compare the difference between HC→ PRCand PRC→ HC. However, each connection itself is a contrast between the recalled-item network and the non-recalleditem network. We simply want to test to see whether the directional connections between these two regions are symmetric.

The first three paragraphs of the Discussion are a mere introduction or repetition of results, and should hence be included in the Introduction or be omitted.

We have fixed the first three paragraphs.

Discussion, fourth paragraph: Why is the decoupling only found in the left hemisphere? This striking difference is completely neglected by the authors. Although I am not too familiar with the cited study by Fell et al. (2006), it does not appear to confirm this difference between hemispheres. As this limitation to the left hemisphere will probably be very obvious to many readers, the authors should discuss it.

We include several citations in the Discussion section to explain this lateralization effect. In particular, the coupling only found in the left hemisphere is in line with neuroanatomical, electrophysiological, and imaging studies that found an association between verbal memory and the left MTL.

Discussion, last paragraph: The authors suggest that fellow researchers can use and extend their method. With this in mind, it would be very beneficial if the authors made their code publicly available.

The code has now been made available on GitHub.

Reviewer #2:[…] The authors do a good job of placing the MSV model in the context of non-parametric methods such as spectral analysis and provide a means to link the two together. Through this linkage the authors are able to show that the correlation between volatility and spectral power is nearly proportional to the analysis frequency in the spectrum. This suggests that neural activity in the gamma band is more volatile than at lower frequencies. This result is of interest because it suggests that the conflicting reports in the literature about the nature and function of gamma band activity in neural activity may in part be due to the application of non-parametric methods that require that the time series be stationary over the data periods of interest.One improvement here would be if the authors also included the analysis of the neural activity during a time window other than the encoding phase. For example, if the countdown phase was also submitted to the same MSV modeling, the hippocampus and perirhinal cortex should show a different functional connectivity than what was reported for successful encoding of words. The authors should also comment on whether the MSV modeling can be applied to a network that includes 20 distinct sites, instead of just 4, as was done in the paper.

As suggested by reviewer 2, we ran a connectivity analysis to compare the connections within the MTL region between the resting state period and the encoding period (including both recalled and non-recalled items). However, we did not find any significant connections after correcting for multiple comparisons (see Author response image 1). This suggests that the contrast between recalled and non-recalled items is perhaps stronger than the contrast between the resting state period and the encoding period.

**Author response image 1. respfig1:** Resting state vs.Encoding Connectivity Contrast.

Reviewer #3:This is excellent work and directs us away from looking at directional measure of influence based on pure autoregressive models (which focus on the conditional mean). Rather interest is focused on the conditional variance, which is shown in a clear way to have advantages over traditional Granger Causality measures. I am convinced by the evidence presented.The paper would benefit by clarifying the sub-types of stochastic volatility models. There is some confusion in the field. The authors seem to be applying a type of GARCH model, sometimes proposed as distinct from SV and sometimes a subtype.I do suggest that some prior work be cited properly. The Wong paper is cited in a way that suggests that it is relevant for emphasizing non-stationary when in reality is also a multivariate stochastic volatility model. The same group even attempted source localization in Galka, Yamashita and Ozaki (2004). This is all well covered in the book by Ozaki on time series modeling of neuroscience data.Finally in the Discussion I would suggest a discussion with combined AR type models and those with stochastic volatility: see Mohamadi et al. (2017).I also do not see if the code will be made publicly available.

We add a literature review of volatility models in the “Volatility is Stochatic” section which includes a discussion of the GARCH-type models. In addition, we cite applications of GARCH models to EEG as suggested by reviewer 3. Furthermore, the MSV approach does incorporate AR-type model with stochastic volatility since we detrend the raw voltage timeseries first using an AR model and then apply the MSV model to the residual timeseries. The code of this study is now available on GitHub.